# Population variability in X-chromosome inactivation across 10 mammalian species

Jonathan M. Werner [1,2], John Hover[1] & Jesse Gillis [1,2] ✉

One of the two X-chromosomes in female mammals is epigenetically silenced in embryonic stem cells by X-chromosome inactivation. This creates a mosaic of cells expressing either the maternal or the paternal X allele. The X-chromosome inactivation ratio, the proportion of inactivated parental alleles, varies widely among individuals, representing the largest instance of epigenetic variability within mammalian populations. While various contributing factors to X-chromosome inactivation variability are recognized, namely stochastic and/or genetic effects, their relative contributions are poorly understood. This is due in part to limited cross-species analysis, making it difficult to distinguish between generalizable or species-specific mechanisms for X-chromosome inactivation ratio variability. To address this gap, we measure X-chromosome inactivation ratios in ten mammalian species (9531 individual samples), ranging from rodents to primates, and compare the strength of stochastic models or genetic factors for explaining X-chromosome inactivation variability. Our results demonstrate the embryonic stochasticity of X-chromosome inactivation is a general explanatory model for population X-chromosome inactivation variability in mammals, while genetic factors play a minor role.

Every female mammalian embryo undergoes X-chromosome inactivation (XCI) as an essential step for successful development[1–3]. XCI evolved to balance the gene dosage between females with two X-chromosomes and males with one X-chromosome[4]. While the exact timing can vary across species[5], XCI usually occurs during pre-implantation embryonic development[6]. During this process, one of the two X-alleles in each female cell is independently, randomly, and permanently chosen for transcriptional silencing to match the single X-allele in male embryos[1,7–9]. The choice of silenced X-allele is inherited through cell divisions, propagating the random choice of allelic inactivation down each cell's subsequent lineage. This produces whole-body mosaicism for allelic X-chromosome expression in each adult mammalian female, originating from very early embryonic development[10].

In humans, both X-alleles are equally likely to be inactivated, but XCI ratios vary widely among adult females, from balanced to highly skewed. XCI ratios affect the phenotypes of X-linked diseases, as they

can either protect or expose individuals to disease variants[10–13]. The factors that influence XCI variability are mostly studied in mice and humans, and include stochasticity[13] and genetics[14–16], but their relative roles are debated[17]. Cross-species comparisons of XCI variability stand to reveal general or species-specific mechanisms of XCI. For instance, genetic determinants of XCI are well-established in lab mice[18–20], but not in humans[17,21,22], where they are more difficult to identify and measure. Exploring XCI variability in other mammals presents the opportunity to test models of stochasticity or genetics in the context of evolution.

Considering first a stochastic model for XCI variability, each cell within an embryo at the time of XCI independently selects an X-allele to inactivate, resulting in ratios of allelic-inactivation varying across embryos purely by chance (Fig. 1A). Closely following Mary Lyon's discovery of XCI in 1961[1], it was recognized that the inherent embryonic stochasticity and permanence of XCI is the simplest explanation for the observed variability in XCI among adults and

[1]Stanley Institute for Cognitive Genomics, Cold Spring Harbor Laboratory, Cold Spring Harbor, NY 11724, USA. [2]Physiology Department and Donnelly Centre for Cellular and Biomolecular Research, University of Toronto, Toronto, ON, Canada. ✉e-mail: jesse.gillis@utoronto.ca

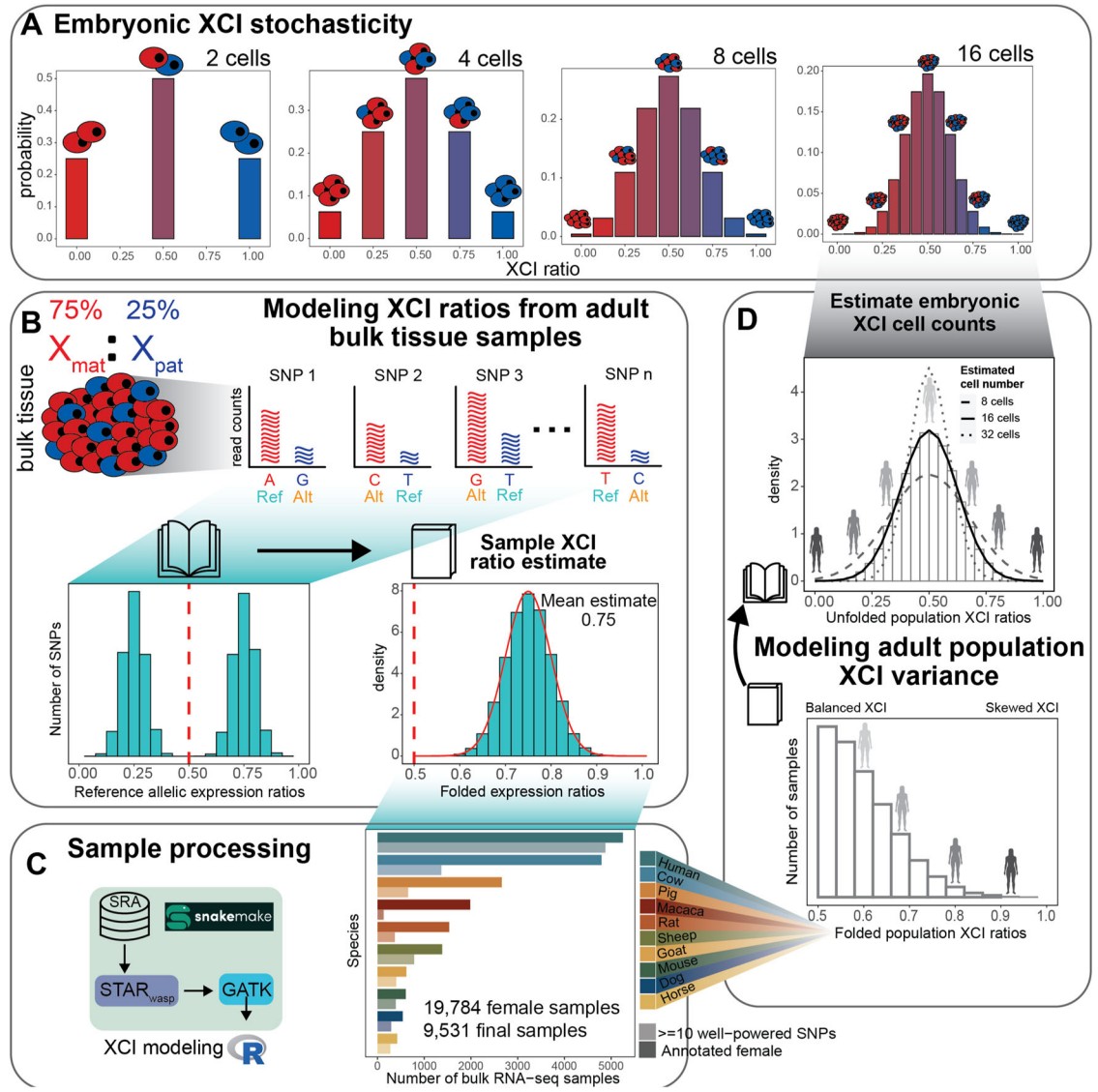

**Fig. 1 | Reference aligned RNA-sequencing data enables scalable modeling of XCI ratios. A** Schematic demonstrating the relationship between the number of cells present at the time of XCI and the probability of all possible XCI ratios. Increased cell numbers result in decreased XCI ratio variance. **B** Schematic for modeling XCI ratios from bulk reference-aligned RNA-seq data. The reference SNPs will contain both maternal and paternal SNPs, representing allelic expression from both parental haplotypes. Folded normal models are fit to the folded reference allelic expression ratios (like folding a book closed), with the mean of the maximum-likelihood distribution as the sample XCI ratio estimate. **C** Schematic for sample processing (genome alignment and variant identification) and a bar graph depicting the number of annotated female samples initially downloaded for each species (bold color), with the number of samples per species with at least 10 well-powered SNPs for XCI ratio modeling after processing (faded color). **D** Schematic demonstrating the population modeling of XCI variability. Folded population distributions are first produced per species and then are unfolded. Normal distributions are fit to the unfolded population distribution to estimate the number of embryonic cells required to produce the observed variance. Details for source data are provided in the Data and Code Availability statements.

positions this adult variability as a window into embryonic events[23–27]. For example, flipping 10 coins is more likely to result in 8 heads than flipping 100 coins is likely to result in 80 heads, meaning that the variability in heads-to-tails ratios depends on the number of coins flipped. Similarly, the variability of XCI ratios in a population of female mammalian embryos is determined by the number of cells at the time of XCI (Fig. 1A). Since each cell inherits its allelic-inactivation from its ancestor, measuring XCI variability in adults can approximate embryonic XCI variability and help infer cell counts at the time of XCI or other early lineage decisions[25,28] (Fig. 1D). Stochastic models have been used to estimate cell counts during embryonic events in human and mice populations for decades[20,23,25,27–29]—but their applicability has not been tested in other mammalian species.

In addition to stochasticity, genetic effects can influence the choice of allelic inactivation and contribute to population variability in XCI ratios. Allelic inactivation during XCI is mediated by the cis-acting long non-coding RNA XIST[30], which silences its corresponding X-allele through epigenetic modifications[31,32]. Heterozygous variants affecting XIST expression can bias allelic inactivation[15]. For example, inbred mice show preferential inactivation of specific X-alleles depending on the parental strains and their corresponding X-chromosome controlling element (XCE) haplotypes[18,20,33]. In humans, genetic influence on XCI is mostly observed in small family studies or disease cases, with no strong evidence for the broad allelic effects seen in mice[21,22]. Another genetic influence on XCI is allelic selection, where disease-associated variants impart a selective effect across the two X-alleles that often results in extreme XCI skew[14,16,34–38]. However, the role of allelic selection outside of a disease context and its relationship to population XCI variability remains to be thoroughly investigated across species. Thus, the relative contributions of stochasticity and genetics to population

XCI variability in mammals remain unclear with currently limited data from mouse and human studies.

In this study, we assess population scale XCI variability and its determinants across ten mammalian species. We source female annotated bulk RNA-sequencing samples from the Sequencing Read Archive (SRA), resulting in a total of 19,784 initial samples derived from 562 individual studies (Fig. 1C), including human samples from the GTEx[39] dataset. Our approach leverages natural genetic variation to sample X-linked heterozygosity and eliminates the requirement for costly phased or strain-specific genetic information to assess XCI ratios across diverse mammals at population scale. We start by establishing the population-level XCI ratio distributions for all ten mammalian species and use models of embryonic stochasticity to predict the number of cells fated for embryonic lineages (Figs. 1D and 2). We then investigate how broad genetic diversity, as indicated by measures of inbreeding (Fig. 3), as well as specific individual variants (Fig. 4), may impact population XCI variability. Overall, our analyses explore how both models of stochasticity and genetic factors can explain population XCI variability across diverse mammalian species.

## Results

### Reference aligned RNA-sequencing data enables scalable modeling of XCI ratios

We use bulk RNA-sequencing (RNA-seq) data to measure the X-linked allelic expression of a sampled tissue by computing allele-specific expression ratios of heterozygous single nucleotide polymorphisms (SNPs). The parental proportion of X-linked allelic reads are expected to follow a binomial distribution dependent on the number of sampled reads and the XCI ratio of the tissue (see methods). The binomial distribution is an appropriate model when the parental identity of sequencing reads is known, which is not the case when aligning to a reference genome. A reference genome will contain SNPs from both parents, making the parental identity of aligned reads ambiguous and producing reference allelic expression ratios that represent expression of both parental X-alleles (Fig. 1B).

Analogous to folding a book on its side closed, we fold the distribution of reference allelic-expression ratios around 0.50 so that values an equal amount above and below 0.5 are in the same bin. This allows us to aggregate data across both alleles and enable a robust estimate of the XCI ratio magnitude for the bulk RNA-seq sample (Fig. 1B). We fit folded-normal distributions to the reference allelic expression ratios of multiple SNPs per sample, which serves as a continuous approximation of the underlying sequencing depth-dependent mixture of folded-binomial distributions per SNP. The mean of the fitted distribution is the estimate of the XCI ratio for the sample (Fig. 1B). We also incorporate specific steps to address confounding factors that can impact measured X-linked allelic expression, namely excluding SNPs with persistent reference bias across samples and chromosomal bins that exhibit probable escape from XCI[40,41] (Supplementary Figs. 1, 2, see methods). Of note, the rat population exhibits a large collection of reference biased SNPs when compared to the other species, likely due to the highly inbred nature of laboratory rat strains. We circumvent this expected issue in the mouse population by leveraging two studies[42,43] that sampled Diversity Outbred (DO)[44] mice, evidenced by the lack of reference-biased SNPs in the mouse population compared to the other species. Additionally, it is important to note our approach detects SNPs present only within RNA molecules, so we will miss variants in non-transcribed proximal regulatory elements, such as the well-described XCE-interval in mice[33]. With regards to escape from XCI, we find the strongest signals of escape near chromosomal ends across all species (Supplementary Fig. 2), suggesting escape within pseudo-autosomal regions is conserved across mammals[40,45]. Previously[28], we validated our SNP filtering and XCI modeling approach using phased RNA-seq data (where haplotype information is known for each variant) from the EN-TEx consortium[46],

achieving nearly perfect agreement in XCI ratio estimates for samples with folded XCI ratios of 0.60 or higher, demonstrating the accuracy of our approach.

By calling SNPs from RNA-seq reads and employing folded distributions to model reference-aligned allelic expression, we can estimate the magnitude of XCI in any female mammalian bulk RNA-seq sample. We source female annotated bulk RNA-seq samples of 9 non-human mammalian species from the SRA database (Fig. 1C), additionally including cross-tissue human samples from the GTEx dataset. As sex annotations were not available on SRA for the two DO mouse studies, we annotate the sex of the mouse samples by thresholding on the total number of reads aligned to the Y-chromosome (Supplementary Fig. 3). After processing, the number of samples with a minimum of 10 well-powered SNPs for estimating XCI ratios are 130 macaca (mean of 28 SNPs ±17 SD), 275 horse (mean of 54 SNPs ±36 SD), 291 dog (mean of 29 SNPs ±13 SD), 369 rat (mean of 28 SNPs ±16 SD), 388 mouse (mean of 87 SNPs ±46 SD), 399 goat (mean of 34 SNPs ±14 SD), 654 pig (mean of 50 SNPs ±28 SD), 784 sheep (mean of 81 SNPs ±43 SD), 1364 cow (mean of 33 SNPs ±19 SD), and 4877 human (mean of 56 SNPs ±23 SD, 314 total individuals) samples (Fig. 1C, Supplementary Fig. 1). Aggregating reference SNP allelic expression ratios for samples with similar estimated XCI ratios (0.05 bins) clearly reveals the expected haplotype expression distributions, demonstrating the applicability of folded models (Supplementary Fig. 4). Following sample-level XCI ratio modeling, we then generate population-level distributions by unfolding the distribution of folded XCI ratio estimates around 0.50, analogous to opening a closed book (Fig. 1D).

As an additional control to ensure the allelic variability we report from X-linked SNPs is specific to XCI, we estimate autosomal allelic imbalances for all samples using the same pipeline and approach as for the X-chromosome analysis (Supplementary Fig. 5, see methods). Comparing allelic imbalances across the two autosomes closest in size to the X-chromosome reveals the vast majority of samples across all species are biallelically balanced for autosomal expression, as expected (Supplementary Fig. 5). Several species (pig, cow, goat, rat, sheep, and dog) exhibit small subsets of samples that are consistently imbalanced across the two autosomes and the X-chromosome, indicative of a global influence on allelic-expression independent of XCI (Supplementary Fig. 5). These samples with global allelic imbalances are excluded from all downstream analysis, ensuring the population distributions of XCI ratios reflect variability specific to XCI.

### Models of embryonic stochasticity explain adult population XCI variability

After generating population distributions of XCI ratios for the 10 mammalian species, we next explore how well models of embryonic stochasticity explain the observed adult XCI ratio variability. The initial variability in XCI ratios among mammalian embryos is dependent on the number of cells present during XCI (Fig. 1A), where adult variability can be modeled to infer embryonic cell counts.

When estimating embryonic cell counts from XCI variability in adult tissues, it is important to note that adult tissues represent only the embryonic lineage of the blastocyst, not the extra-embryonic lineages. This positions XCI variability of adult tissue samples as informative for the number of cells present within the last common lineage decision for all adult cells, i.e. the number of cells present within the epiblast of the mammalian blastocyst. If XCI occurs after epiblast specification, XCI ratio variability is determined by the number of epiblast cells at the time of XCI. If XCI occurs before epiblast specification, the variability is influenced by both the initial stochasticity of XCI and the stochasticity of cell sampling during epiblast lineage specification. Without cross-tissue sampling of both extra-embryonic and embryonic tissues, the temporal ordering of XCI among these lineage events cannot be resolved. Therefore, estimating cell counts based solely on XCI variability in adult tissues provides an

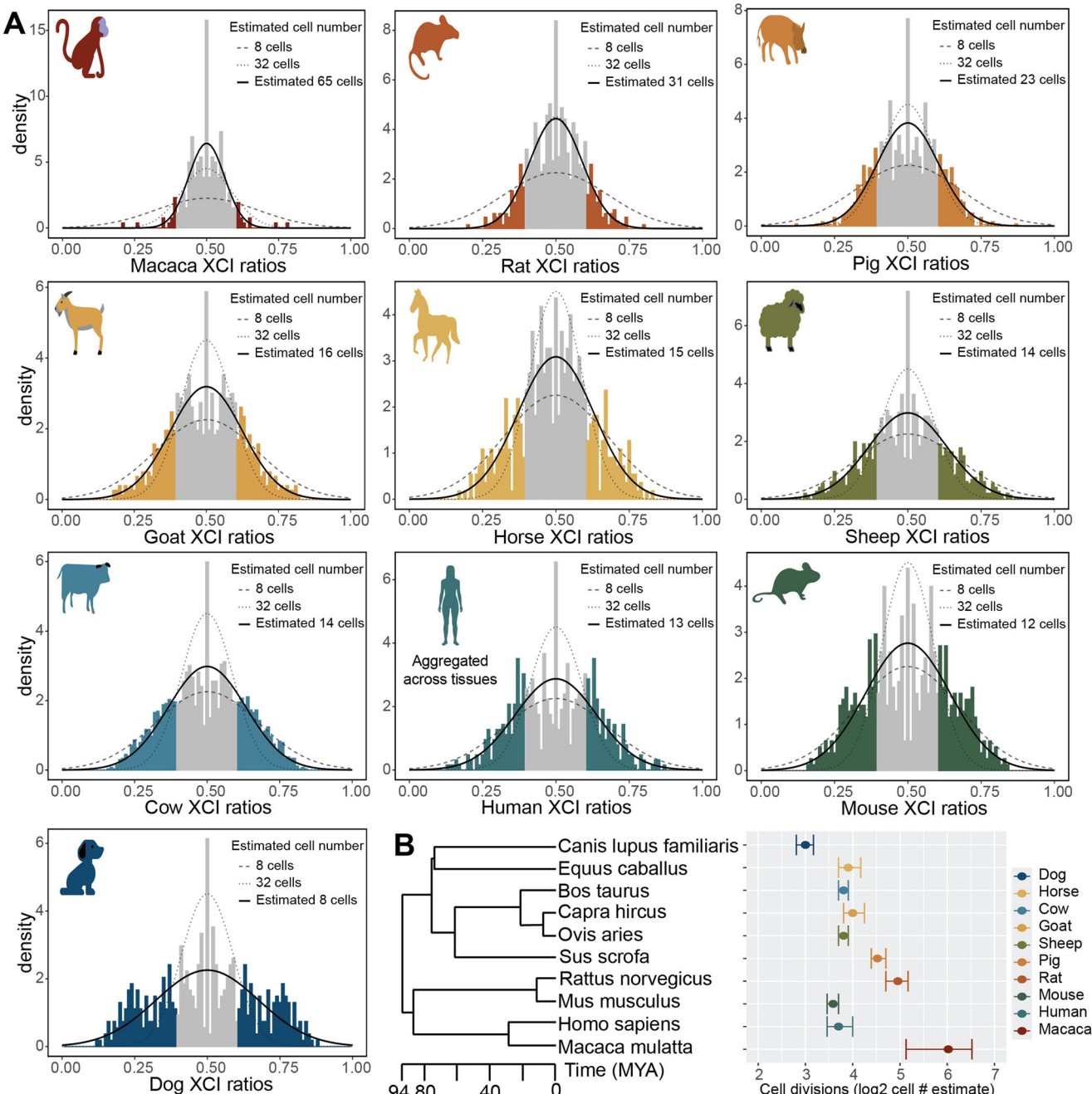

**Fig. 2 | Models of embryonic stochasticity explain adult population XCI variability. A** Unfolded distributions of XCI ratios per species, with the maximum-likelihood normal distribution depicted in bold, fitted to the tails of the distributions (shaded in sections of the distributions, unfolded estimated ≤0.40 and ≥0.60). **B** Phylogenetic tree of the sampled mammalian species with their estimated embryonic cell counts on a log-2 scale, depicting the number of cell divisions that separate the estimated cell counts between the species. Error bars are 95% confidence intervals computed through bootstrap simulations with $n = 2000$, with the measure of center corresponding to the estimated cell number per species on a log-2 scale, denoted in each species' panel in **A** as Estimated # cells. Details for source data are provided in the Data and Code Availability statements.

estimate of the number of cells present in the epiblast at the time of XCI.

Figure 2A presents the unfolded population distributions of XCI ratios in the 10 mammalian species we sampled, ranging from the least variable (macaca) to most variable (dog). We fit normal distributions as continuous approximations to the underlying binomial distribution that defines the relationship between cell counts and XCI ratio variability (Fig. 1A, D, see methods). We focus on the tails of the distributions for our model fitting (colored in portions of the distributions, unfolded estimates ≤0.40 and ≥0.60, Fig. 2A), for two reasons. Our analysis of autosomal allelic ratios (Supplementary Fig. 5) highlights

that samples with no expected allelic imbalance produce folded skew estimates that vary between 0.5 and 0.6 and our previous work[28] using phased data indicated model misspecification around the point of folding (0.50). Fitting to the tails of the empirical distribution is therefore a more accurate representation of variability specific to XCI.

At a broad level, population XCI ratio variability varies substantially across the sampled mammalian species. Our estimates for the number of epiblast cells present at the time of XCI include 65 (macaca), 31 (rat), 23 (pig), 16 (goat), 15 (horse), 14 (sheep), 14 (cow), 13 (human), 12 (mouse), and 8 (dog) cells, with associated 95% confidence intervals presented in Fig. 2B. Importantly, species with similar

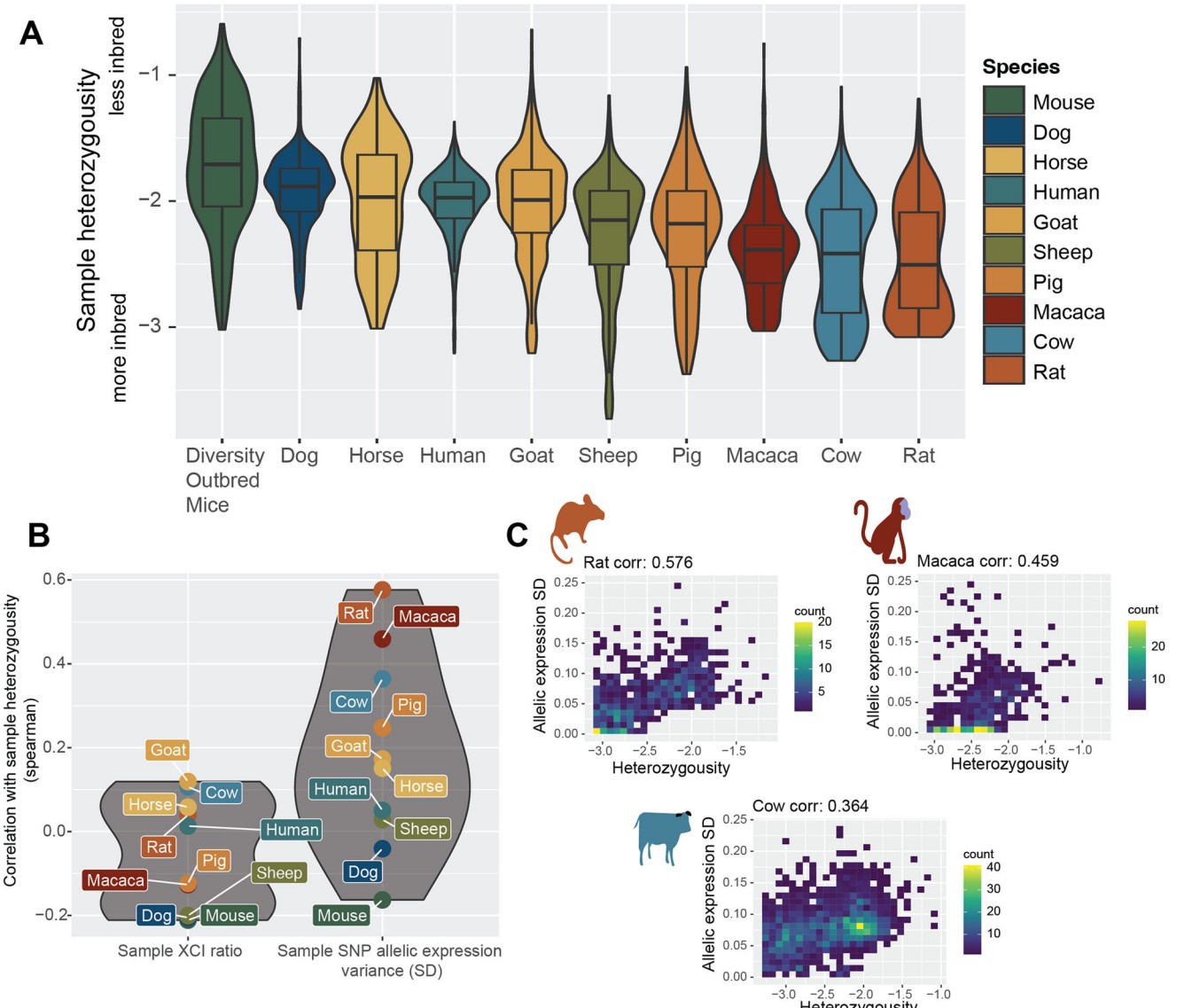

**Fig. 3 | XCI ratios are not associated with X-linked heterozygosity.**
**A** Distributions of sample X-linked heterozygosity per species ordered by the median value. The y-axis is in log-10 scale, depicting the ratio of SNPs per sample to all unique identified SNPs per species. Box plots indicate median (middle line), 25th, 75th percentile (box) and 1.5 times the inter-quartile range from the first and third quartiles (whiskers) with sample numbers per species as follows: macaca $n = 130$, horse $n = 275$, dog $n = 291$, rat $n = 369$, mouse $n = 388$, goat $n = 399$, pig $n = 654$, sheep $n = 784$, cow $n = 1364$, human $n = 4877$. **B** The spearman correlation coefficients between sample X-linked heterozygosity and either the estimated standard deviation (SD) in X-linked allelic expression or the estimated XCI ratio of the sample (the SD and mean of the maximum-likelihood folded-normal model per sample). **C** 2D Scatter plots of sample heterozygosity compared to the sample estimated X-linked allelic expression SD for the three species with moderate correlation coefficients. Color bars represent the number of samples in each 2D bin. Plots for the other species are in Supplementary Fig. 7. Details for source data are provided in the Data and Code Availability statements.

numbers of detected SNPs per sample and total sample size exhibit variable cell number estimates, indicating it is unlikely our estimates are driven by technical effects across the species (Supplementary Fig. 6). We additionally down sample all species to the smallest sample size present (130 macaca samples) and achieve virtually identical cell number estimates, demonstrating variation in cell number estimates across species is not driven by sample size differences (Supplementary Fig. 6). The error between the empirical XCI ratio distributions and the normal fitted distributions is strikingly small, with a mean of 0.00588 sum-squared error (±0.00965 SD) across the species (Supplementary Fig. 6). This shows models of embryonic stochasticity can explain observed XCI ratio variability in adult populations exceptionally well.

For the least and most variable species (macaca and dog), the estimated autosomal imbalances offer additional context for the reported XCI population variability. The reported X-linked

variability in macaca is in excess to the reported autosomal allelic variability, which itself is highly consistent across species (Supplementary Fig. 5). This demonstrates the X-linked population variability for macaca, while strikingly small, still varies beyond the extremely consistent autosomal variability present across species and is specific to the X-chromosome, representing informative variability for estimating cell counts. On the other hand, the dog population is the only one that contains samples with strong allelic imbalances on only one autosome, where autosomal imbalances in all other species are global (Supplementary Fig. 5). This is suggestive of broader genomic incompatibilities within the dog population. The reported X-linked population variability in dog is likely a combination of XCI and broader allelic incompatibilities, positioning our estimate of 8 cells as a likely underestimate due to excess variability outside of XCI.

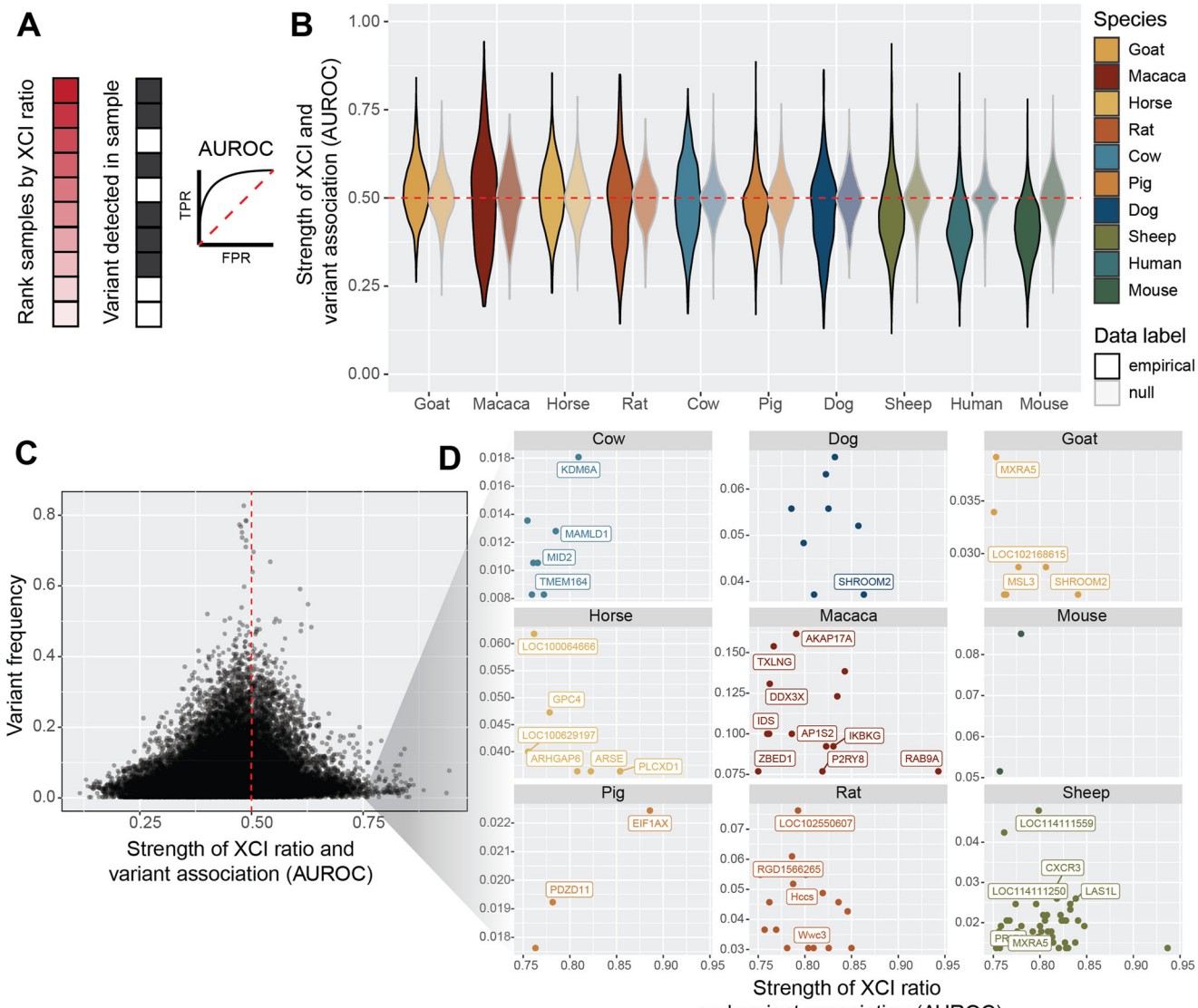

**Fig. 4 | Low frequency variants exhibit moderate associations with XCI ratios.**
**A** Schematic depicting the AUROC quantification for testing the association between individual variants and extreme XCI ratios. Samples are ranked by their estimated XCI ratio, with the dark shaded red squares representing samples with more extreme XCI ratios. The position of samples with a given individual variant (gray squares) within the ranked list is used to compute the AUROC statistic. A variant with an AUROC value of 1 means all samples with that variant were at the top of the ranked list, whereas an AUROC value of 0.5 represents a random ordering of samples within the ranked list. **B** Distributions of variant AUROCs for each species compared to a species-specific null distribution of AUROC values (faded distributions, see methods), ordered by the mean value of the empirical distributions. The

red dotted line depicts an AUROC of 0.50, performance due to random chance.
**C** Scatter plot of variant AUROCs compared to each variant's prevalence (percent of samples with that variant, relative for each species) for all variants across all species. The red dotted line depicts an AUROC of 0.50, performance due to random chance. A threshold of AUROC > = 0.75 was used to identify SNPs with moderate associations with XCI ratios. **D** Scatter plots depicting the same information as in C for the variants with moderate associations with XCI ratios, but split by each species and including gene annotations. SNPs not within annotated genes are unlabeled. Gene labels not present due to overlapping labels are Sheep: *IL3RA, LOC101108113, LOC101115509, LOC101117055, LOC105605313, LOC121818231, PPP2R3B, PRKX*).
Details for source data are provided in the Data and Code Availability statements.

Modeling XCI ratio variability across numerous species allows comparisons in light of evolution for determining generalizable or species-specific characteristics in XCI. Broadly, we demonstrate XCI ratios are variable in each species we assess, revealing variability in XCI ratios itself as a conserved characteristic of XCI. The exact variance in XCI ratios varies across the species, with differences in the timing of XCI and/or differences in cell counts for embryonic/extra-embryonic lineage specification as two putative explanations. We compare our estimated cell counts to the evolutionary relationships among the species we assess (Fig. 2B), suggesting that variability in these early embryonic events can be recent evolutionary adaptations. This is highlighted by the large differences in cell counts between macaca and humans, as well as between rats and mice. When viewed through the

lens of cell divisions (log2 of the estimated cell counts, Fig. 2B), the differences in XCI ratio variability among the species can be explained by differences in a range of only 3 cell divisions, a narrow developmental window. This demonstrates even slight changes in the timing of XCI or cell counts for embryonic/extra-embryonic lineage specification across mammalian species can produce large differences in population XCI ratio variability, as explained through the inherent stochasticity of XCI.

## XCI ratios are not associated with X-linked heterozygosity
After determining stochastic models can explain population XCI ratio variability across mammalian species, we turn to testing whether we can identify any genetic correlates with XCI ratios. Our approach

leveraging natural genetic variation to quantify XCI ratios enables us to assess a large catalog of genetic variants for associations with XCI ratios across mammalian species (10,735 macaca SNPs, 12,024 rat SNPs, 28,339 mouse SNPS, 23,603 pig SNPs, 16,123 goat SNPs, 10,281 horse SNPs, 53,505 sheep SNPs, 18,509 cow SNPs, 16,168 human SNPs, and 10,050 dog SNPs). One putative genetic contribution to XCI ratio variability is allelic selection during development, where increased X-linked heterozygosity (i.e., genetic distance), is more likely to produce selective pressures between the two X-alleles. It follows that samples with higher X-linked heterozygosity would be expected to exhibit more extreme XCI ratios.

We score X-linked heterozygosity per sample as the ratio of the detected SNPs within a sample to the number of unique SNPs identified across all samples, relative for each species (Fig. 3A). This quantification also serves as a measure of inbreeding, with decreased heterozygosity associated with a higher degree of inbreeding[47]. The trend in heterozygosity across species is as expected, with rats (likely laboratory strains) as the most inbred (Fig. 3A). Next, we examine the correlations between sample heterozygosity and the estimated XCI ratio, as well as the estimated allelic variability across SNPs in each sample (mean and standard deviation of the fitted folded-normal distribution per sample, Fig. 3B). Across all species, X-linked heterozygosity shows a near-zero correlation with the estimated XCI ratio, indicating a lack of association between X-linked genetic heterozygosity and XCI ratio variability (Fig. 3B). However, we observe moderate correlations between sample heterozygosity and the estimated variability in SNP allelic ratios in three species: rat (corr: 0.576), macaca (corr: 0.459), and cow (corr: 0.364), notably the most inbred species (Fig. 3A, Supplementary Fig. 7). The increased variability in allelic expression present only within the most inbred species could potentially reflect gene-specific regulatory events between parental haplotypes[48] rather than a direct genetic effect on XCI.

## Low frequency variants exhibit moderate associations with XCI ratios

After investigating relationships between genetic variation and XCI ratios at a broad level across the whole X-chromosome, we next asked if individual variants might be associated with extreme XCI ratios. Variants that affect the expression and/or function of the genetic elements that control XCI can result in highly skewed XCI ratios, as documented in human studies[15]. This can also occur in other X-linked genes, if the resulting differential in gene activity exerts a selective pressure across the X-alleles, as documented in disease cases[14,16]. We test the association between XCI ratios and individual variants for all variants detected in each species with a minimum of 10 samples, quantified through the area-under-the-receiver-operating-curve statistic (AUROC). For each species, we rank the samples based on their estimated XCI ratio and score the placement of samples carrying a given variant within the ranked list (Fig. 4A). If all the samples with that variant are at the top of the ranked list, the XCI ratio can be said to have perfectly predicted the presence of that variant, quantified with an AUROC of exactly 1. An AUROC of 0.50 indicates the XCI ratio performs no better than random chance for predicting the presence of the variant.

The distribution of AUROCs for each species show striking similarities to a null comparison (Fig. 4B, see methods), indicating a pervasive lack of association between XCI ratios and individual variants. However, a small subset of variants in each species exhibits moderate associations (AUROCs ≥0.75 and FDR-corrected $p$-value ≤0.05). By comparing each variant's AUROC with its frequency in the species, we find that the variants with moderate associations occur at low frequencies within the sampled populations (Fig. 4C, Supplementary Fig. 8). We investigate whether this relationship is simply due to a lack in power with bootstrap simulations, demonstrating moderate AUROCs (≥0.75) are robust to their small sample sizes

(Supplementary Fig. 8). Figure 4D displays these variants along with their gene annotations for each species. Notably, we observe no statistically significant variant-XCI ratio associations in the GTEx human population when performing either a tissue-specific or donor-specific analysis, as well as only considering the sample per donor with the highest sequencing depth (Supplementary Fig. 9). While the GTEx dataset is comprised of thousands of tissue samples, only 314 female individuals are present in our final dataset. We test the effect of a small population size by down sampling the cow data to 300 samples and scoring variant-XCI ratio associations (Supplementary Fig. 9). All of the cow variants that we originally identified as significantly associated with XCI ratios are no longer detected in the down sampled data, in line with the observation that variants with associations to XCI ratios occur at low frequencies within mammalian populations. Increased population sampling is likely required to identify further genetic associations with XCI ratios.

Several genes with moderate AUROCs have prior evidence for escaping XCI in humans[49], bringing into question their associations with extreme XCI ratios in our analysis. To explore further, we compare the estimated XCI ratios of samples to the allelic ratios of all detected variants for genes with at least one variant significantly associated with XCI ratios. We report several examples across species where the allelic expression of individual variants from these putative escape genes does in fact exhibit the expected balanced biallelic expression of escape from XCI while also being enriched in samples with increased XCI ratios (Supplementary Fig. 9). A gene that escapes XCI will be biallelically expressed; this suggests the variant-specific association we detect within these XCI escapers likely reflects a haplotype-effect, where the variant is linked to a haplotype influencing XCI ratios, rather than an effect from the gene/variant itself. Further analysis with phased data to assess potential haplotype effects may help identify genetic associations with XCI ratios. Overall, our assessments of chromosome-wide genetic variability and individual variants do not reveal genetic associations robust enough to explain population XCI ratio variability across all 10 mammalian species.

## Putative mouse XCE-*Xist*-haplotypes exhibit highly variable XCI ratios

One of the most well-documented instances of a genetic association with XCI ratios is the XCE-haplotypes in laboratory mouse strains, where a preferential ordering of allelic inactivation exists across haplotypes[33]. The DO mice we utilize are expected to be genetically diverse combinations of various lab strains and it is highly likely a mix of XCE-haplotypes are present within this population and may have an impact on XCI ratios. Such a haplotype-specific effect would be missed in our previous AUROC variant-specific analysis of XCI ratios. Since we only sample variants present within RNA molecules and the XCE-interval is a proximal non-transcribed regulatory element of *Xist*[33], we reason variants present within *Xist* are likely linked to XCE-haplotypes and may be informative for identifying putative XCE-*Xist*-haplotypes. We identify 4 putative XCE-*Xist*-haplotypes as determined by groups of samples with shared *Xist* variants (Supplementary Fig. 10). As a general observation across haplotypes and the two studies we sample from, XCI ratios of samples with the same haplotype are highly variable (Supplementary Fig. 10), suggesting XCI ratios are not definitively determined by *Xist* genotypes within the DO mice population. The haplotype with the seemingly largest effect, evidenced by 2 samples with highly skewed XCI ratios (0.799 and 0.782) in the one study that collected striatal tissue, conversely exhibits XCI ratios ranging from balanced to moderately skewed in the second study, which collected pancreatic tissue (Supplementary Fig. 10). While this may be indicative of a tissue-specific effect of a particular XCE-*Xist*-haplotype, far greater sample sizes with higher genetic resolution to confirm haplotypes are needed for validation. In general, the variability in XCI ratios within

putative XCE-*Xist*-haplotypes suggests non-genetic contributions to XCI variability of DO mice.

## Discussion

We modeled tissue XCI ratios from bulk RNA-seq samples across 10 mammalian species and revealed population-level variation in XCI ratios that likely reflects differences in developmental events such as XCI timing or lineage specification. We showed that models of embryonic stochasticity fit the XCI data exceptionally well and estimated epiblast cell counts at the time of XCI across species. We also searched for genetic factors influencing XCI ratios and found a pervasive lack of strong genetic associations with XCI ratios, indicating that population XCI variability is better explained by the inherent stochastic nature of XCI rather than through genetic mechanisms.

The lack of cross-mammalian comparisons of population XCI variability has previously limited our understanding on the sources of XCI variability in mammals. The existence of XCE-haplotypes in laboratory mice[18–20,33] has supported the hypothesis that a similar genetic mechanism can exist in humans and drive population XCI variability[21], though evidence for XCE-haplotypes in human populations remains inconclusive[22] and data from other mammalian species is historically absent. Although genetic influences on XCI, particularly variants affecting *XIST*[15] or disease-associated variants[34–37], have been identified, they do not constitute a general mechanism that can fully account for observed population-level XCI variability across species. In particular, allelic selection via genetic variability across the X-alleles has been put forth as an explanatory mechanism for XCI ratio variability[14,16], but is almost exclusively studied in a disease context and typically associated with extreme XCI ratios[14,16,17,34–38], which is conflicting with the continuous population variability we report across species. Our measures of X-linked heterozygosity have near-zero association with XCI ratios in all 10 mammals we assessed, a strong indication that genetic variability on the X-chromosome has little influence on XCI ratios outside of disease. This is supported by the observation of depleted X-chromosome genetic variability via strong rates of purifying selection[50–53], rendering both parental alleles as largely equivalent at population scales. Our approach for extracting heterozygous variants from RNA-seq data[28], while providing a sample of genetic variability, is still able to assess hundreds of X-linked genes and chromosome-wide heterozygosity per species for associations with XCI and culminated in only weak evidence of limited genetic influence on XCI ratios. In contrast, we demonstrated models of embryonic stochasticity can explain population XCI variability with exceedingly small amounts of error consistently across mammalian species, providing a much more general explanation for population XCI variability.

Besides X-linked disorders and *XIST*-variants, other factors that may affect XCI ratio variability are genomic incompatibilities[48] and stochastic allelic drift during development[20] and/or aging as in the well-reported case of increased skewing of blood samples with age[29,54]. We found an association between the variance in X-linked allelic expression and the degree of inbreeding for some species (Fig. 2B), as well as autosome-specific allelic imbalances in dog samples (Supplementary Fig. 5). This implies that X-linked allelic expression variability may result from both the bulk tissue XCI ratio and the genomic incompatibilities between the parental genomes[48], depending on the species. We controlled for global allelic imbalances by excluding samples that showed consistent autosomal imbalances (Supplementary Fig. 5), which confirms the allelic-expression variability on the X-chromosome is specific to XCI. Turning to allelic drift, developmental allelic drift may introduce XCI ratio variability beyond the initial random choice of allelic inactivation[20]. While our previous cross-tissue analysis of XCI ratios in humans[28] showed consistent XCI ratios across tissues, suggesting allelic drift is not a major factor in XCI ratio variability, similar data for non-human mammals is missing. In general, we

cannot account for tissue-specific or age-related effects in this dataset as these sample annotations are almost universally absent for the SRA sourced data. These factors indicate that our epiblast cell count estimates are lower bound estimates for the number of cells needed to produce the observed XCI ratio variability as purely derived from embryonic stochasticity. Our statistical modeling approach here would be greatly complemented by future experimental validation of the timing and cell counts present during XCI across species.

Regarding the timing of XCI and our epiblast cell count estimates, exploring known temporal variability in XCI across species provides additional context. In mice, random XCI occurs within the epiblast soon after its specification and is readily identified by monoallelic expression of *XIST*[55]. In macacas and humans, inactivation appears more continuous; data show progressive chromosome-wide silencing over several days, with a shift from biallelic to monoallelic *XIST* expression[56–58]. This lengthy continuous inactivation obscures the exact timing of XCI in these species and highlights that XCI hallmarks in mice, namely rapid mono-allelic *XIST* expression, are not readily applicable to other species; indeed, many species initially exhibit biallelic *XIST* expression[59]. In context, our epiblast cell counts estimate the number of cells involved in XCI, not the exact timing. For instance, our macaca and human cell counts indicate approximately four times as many cells are present within the epiblast at the time of XCI in macacas compared to humans. This difference could result from delayed XCI or a greater number of cells fated for the epiblast in macacas compared to humans. In general, our cell count estimates reflect population sizes at the time of XCI and we attribute variability in cell counts as most likely due to differences in XCI timing or lineage specification dynamics across species.

An important caveat to our analyses of genetic influences on XCI ratios is that we are limited to assessing variants present within RNA molecules, which are necessary for quantifying allele-specific expression. Consequently, we likely miss many non-transcribed regulatory variants that may significantly influence XCI ratios. The XCE-interval in mice is one example, a proximal regulatory element to *Xist* known to influence XCI ratios in heterozygous lab crosses[33]. We identified putative XCE-haplotypes in the DO mice population using *Xist* variants as proxies and showed XCI ratios are highly variable among samples sharing a haplotype, indicating other factors outside of *Xist* genotypes/putative XCE-haplotypes influence XCI ratios in DO mice. While this suggests the effects of XCE-haplotypes in DO mice are minor and more pronounced in inbred lab crosses, we cannot exclude the possibility of similar genetic influences on XCI ratios in non-transcribed regions among our sampled species. Our approach trades comprehensive genetic screening for scalability, enabling the assessment of XCI ratios across thousands of samples from different species. While we demonstrate models of embryonic stochasticity explain observed population XCI ratio variability far better than genetic associations, we are limited in the type of genetic variability we can assess. Importantly, we do not discount the well-documented role of genetics in XCI ratios for rare and exceptional cases (e.g., disease); instead, we advance embryonic stochasticity as the parsimonious explanation of XCI variability in normal populations.

## Methods

### Snakemake pipeline for RNA-seq alignment and variant identification

All non-human mammalian fastq data was downloaded from the Sequencing Read Archive (SRA, https://www.ncbi.nlm.nih.gov/sra), where only samples annotated as female were selected, using the metadata provided through SRA. We sourced Diversity Outbred mice data from two studies[42,43] where sex annotations were not available on SRA and identified female samples as those with less than 200 CPM counts aligned to the Y-chromosome (Supplementary Figure 3). Details for download and processing of the GTEx[39] data can be found

here[28]. The entire sample processing pipeline uses a standard collection of bioinformatics software tools, all available for installation via Conda (STAR[60] v2.7.9a, GATK[61] v4.2.2.0, samtools[62] v1.13, igvtools[63] v2.5.3, and sra-tools 2.11.0). All Snakemake workflow rules, environment setup procedure, analysis commands and options, and underlying libraries are available on Github at https://github.com/gillislab/cross_mammal_xci, and https://github.com/gillislab/xskew. Briefly, a.fastq file acts as input, for either single- or pair-end sequencing experiments, and a.vcf and.wig file are produced as outputs for subsequent compiling of allele-specific read counts in R v4.3.0. The R script used for combining the.vcf and.wig information is also made available at https://github.com/gillislab/cross_mammal_xci/tree/main/R. Genome generation and alignment was performed with STAR, with the addition of the WASP[64] algorithm for identifying and excluding reference biased reads. We extract chromosome-specific alignments from the.bam file (X chromosome or specific autosomes) and use GATK tools to identify heterozygous SNPs from that chromosome. The suite of GATK tools for identifying heterozygous variants from RNA-sequencing data was used following the GATK Best Practices recommendations. Specifically, the tools utilized include AddOrReplaceReadGroups -> MarkDuplicates -> SplitNCigarReads -> HaplotypeCaller -> SelectVariants -> VariantFiltration.

Reference genomes and gene annotations (.gtf files) for each species were sourced from the NCBI Refseq database (https://www.ncbi.nlm.nih.gov/refseq/). In each case the latest assembly version path was used, and the genomic.fna and genomic.gtf was downloaded. Annotated and indexed genomes were generated with STAR using –runMode genomeGenerate with default parameters.

## SNP filtering
Only SNPs with exactly two identified genotypes were included for analysis and indels were excluded. We required each SNP to have a minimum of 10 reads mapped to both alleles for a minimum read depth of 20 reads per SNP. Gene annotations for all SNPs were extracted from the species-specific.gtf files. For XCI ratio modeling, we only used SNPs found within annotated genes. For any sample with multiple SNPs identified in a gene, we took the SNP with the highest read count to be the max-powered representative of that gene, so each individual SNP is representative of a single gene. In addition to implementing the WASP algorithm for excluding reference biased reads, we filter out SNPs within each species whose mean expression ratios across samples deviate strongly from 0.50 (mean allelic ratio <0.40 and >0.60, Supplementary Fig. 1). This SNP filtering also excludes potential eQTL effects that may impact allelic-expression outside of the underlying XCI ratio.

## Identifying and excluding chromosomal regions that escape XCI
We reasoned robust escape from XCI would produce more balanced biallelic expression in samples with skewed XCI. We performed an initial pass at XCI ratio modeling including all well-powered SNPs in a sample to identify samples with skewed XCI ratios (XCI ratios ≥0.70 for all species except rat and macaca, where a threshold of 0.60 was used due to a reduced incidence of skewed XCI in these species). Using the subset of skewed samples for each species, we averaged the folded allelic-expression ratios for all SNPs present in 1 mega-base (MB) bins across the X-chromosome (Supplementary Fig. 2). Chromosomal-bins that displayed balanced allelic expression in opposition to the clearly skewed allelic expression of the rest of the chromosome were excluded from analysis. Specifically, chromosomal bins with an average allelic-expression <0.65 for pig, goat, horse, sheep, mouse, and cow, <0.60 in rat and macaca, and <0.675 in dog were excluded (Supplementary Fig. 2) The ends of the X-chromosome in all species, except rat and mouse, demonstrated strong balanced biallelic expression, indicative of escape within putative pseudo-autosomal regions. We excluded any bin within these putative pseudo-autosomal regions

regardless of average allelic expression. The escape threshold for dog was increased to exclude all bins within the dog putative pseudo-autosomal region.

## Modeling XCI ratios with the folded-normal distribution
Starting with a single parental allele, the sampled maternal allelic-expression of a heterozygous X-linked SNP can be modeled with a binomial distribution, dependent on the ratio of active maternal X-alleles in the sample and the read depth of the SNP.

$$\frac{X_{mat}}{n_{reads}} \sim \frac{Bin(n_{reads}, p_{mat})}{n_{reads}}; E\left[\frac{X_{mat}}{n_{reads}}\right] = p_{mat}; Var\left(\frac{X_{mat}}{n_{reads}}\right) = \frac{p_{mat}(1-p_{mat})}{n_{reads}}, \quad (1)$$

where $X_{mat}$ is the number of maternal allelic reads, $n_{reads}$ is the read depth of the SNP, and $p_{mat}$ is the ratio of active maternal X-alleles. When aligned to a reference genome, the parental phasing information is lost and the allelic-expression of X-linked SNPs can instead be modeled with the folded-binomial model[65,66]. Since SNPs vary in read-depth, we use a folded-normal model as an approximation of the underlying mixture of sequencing depth-dependent folded-binomial distributions. The probability of allelic-expression under the folded-normal model is defined as:

$$\Pr(x_{ratio}; \mu, \sigma^2) = \frac{1}{\sqrt{2\pi}\sigma}e^{-\frac{(x_{ratio}-\mu)^2}{2\sigma^2}} + \frac{1}{\sqrt{2\pi}\sigma}e^{-\frac{(x_{ratio}+\mu-1)^2}{2\sigma^2}}, \text{for } \mu \in [0.50, 1],$$
$$(2)$$

where $x_{ratio}$ is the folded allelic-expression ratio of a SNP, $\mu$ is the folded XCI ratio of the sample, and $\sigma$ is the standard deviation of the folded-normal distribution. We utilize a maximum-likelihood approach (negative log-likelihood minimization of Eq. (2)) to fit folded-normal distributions to the observed folded allelic-expression ratios of at least 10 filtered SNPs per sample, taking the $\mu$ parameter of the maximum-likelihood folded-normal distribution as the folded XCI ratio estimate of the sample.

## Modeling autosomal imbalances
The folded-normal model can also be applied to autosomal data to estimate allelic-imbalances. For each species, we extract chromosome-specific alignments from the.bam file for the two autosomes closest in size to the X-chromosome (Supplementary Fig. 5). We employ the exact same processing pipeline and thresholds as used for the X-chromosome. Any sample that displayed an autosomal imbalance greater than or equal to a folded estimate of 0.60 (dotted lines in Supplementary Fig. 5A) on either autosome was excluded from downstream analysis.

## Modeling population XCI variability with models of embryonic stochasticity
XCI is a binomial sampling event, where the number of cells choosing to inactivate the same X-allele follows a binomial distribution defined as:

$$X \sim Bin(n_{cells}, p_{inact}), \quad (3)$$

where $X$ is the number of cells inactivating the same X-allele, $n_{cells}$ is the number of cells present at the time of XCI, and $p_{inact}$ is the probability of inactivation (0.50).

Embryonic XCI ratios can be modeled as:

$$\frac{X}{n_{cells}} \sim \frac{Bin(n_{cells}, p_{inact})}{n_{cells}} \quad (4)$$

We estimate $n_{cells}$ by fitting normal distributions to the unfolded population XCI ratio distributions of each species, as a continuous

approximation for the underlying binomial distribution. The variance of the normal distribution is defined as:

$$\text{var}_{normal} = Var\left(\frac{Bin(n_{cells}, p_{inact})}{n_{cells}}\right) = \frac{p_{inact}(1 - p_{inact})}{n_{cells}} = \frac{.5(1 - .5)}{n_{cells}} \quad (5)$$

We model population XCI ratios as:

$$\frac{X}{n_{cells}} \sim Norm(\mu, \sqrt{\text{var}_{normal}}), \quad (6)$$

where $\mu = p_{inact} = 0.50$ and $\text{var}_{normal}$ is computed for $n_{cells} \in [2, 200]$. We identify the normal distribution with minimum sum-squared error between its CDF and the empirical population XCI ratio CDF, minimizing error over the tails of the distributions with percentiles ≤0.40 or ≥0.60 (Supplementary Fig. 6). We compute 95% confidence intervals about the cell number estimate $n_{cells}$ through bootstrap simulations. We sample with replacement from the empirical population XCI ratio distribution, matching the sample size of the original empirical population distribution, and fit a normal model to derive a bootstrap estimate of $n_{cells}$. We repeat this for 2000 simulations to generate a bootstrapped distribution of $n_{cells}$, from which we derive the 95% confidence intervals, defined as the interval where 2.5% of the bootstrapped distribution lies outside either end.

We down sample the population XCI ratio distribution to 130 for each species to match the sample size of macaca, the species with the smallest sample size. We sample with replacement and then estimate cell numbers as previously described, repeating for 2000 simulations. The mean cell number estimate and 95% confidence intervals for each down sampled species is reported in Supplementary Fig. 6D.

### Measuring sample X-linked heterozygosity

We compute sample heterozygosity as the ratio of SNPs detected in a sample (20 read minimum) to the total number of unique SNPs identified across all samples for a given species. We quantify associations between X-linked heterozygosity and XCI ratios as the spearman correlation coefficient between the sample X-linked heterozygosity ratio and the fitted mean and variance of the maximum-likelihood folded-normal distribution of the sample (Fig. 3B, C, Supplementary Fig. 7). We only consider samples with at least 10 detected SNPs.

### Quantifying variant associations with extreme XCI ratios

We quantify the strength of XCI ratios as a predictor for the presence of a given variant through the AUROC metric. Given a ranked list of data (XCI ratios) and an indicator of true positives (samples with a given variant), the AUROC quantifies the probability a true positive is ranked above a true negative. An AUROC of 1 indicates all true positive samples were ranked above all true negative samples, demonstrating XCI ratios were a perfect predictor for the presence of that variant. An AUROC of 0.50 indicates random placement of true positives and negatives in the ranked list, demonstrating XCI ratios performed no better than random chance for predicting the presence of that variant. We compute the AUROC through the Mann–Whitney U-test, defined as:

$$AUROC = \frac{U}{n_{pos} + n_{neg}}, \quad (7)$$

where $U$ is the Mann–Whitney U-test test statistic, computed in R with wilcox.test(alternative = 'two.sided'), $n_{pos}$ is the number of true positive samples and $n_{neg}$ is the number of true negative samples. We generate a null AUROC per variant by randomly shuffling the true positive and negative labels. The variant frequency is defined as the number of samples that carry a given variant over the total number of samples for a given species. The $p$-value for a given AUROC is the $p$-value associated with the Mann–Whitney U-test test statistic ($U$),

where we determine significance as an FDR-corrected $p$-value ≤0.05. We perform FDR correction for all $p$-values computed for all variants across the 10 species through the Benjamini–Hochberg method, implemented in R via p.adjust(method ='BH').

We estimate the power of each variant through bootstrap simulations. We randomly sample with replacement the XCI ratios of the true positive and true negative samples, those that either carry or do not carry a given variant. We match the sample size of the original true positive and negative labels. We compute a bootstrapped AUROC and $p$-value from the simulated data, repeating for 2000 simulations to compute a bootstrapped distribution of AUROCs. The AUROC power (Supplementary Fig. 8B) is defined as the fraction of bootstrapped AUROCs that are significant, using a significance threshold of FDR-corrected $p$-value ≤0.05. The AUROC effect size power (Supplementary Fig. 8C) is defined as the fraction of bootstrapped AUROCs that are ≥0.75. We also report the variance of the bootstrapped AUROC distribution per variant in Supplementary Fig. 8D. We exclude all variants classified as reference biased from Supplementary Fig. 1, with the distributions of AUROCs for the reference biased and non-reference biased SNPs presented in Supplementary Fig. 8E.

We assess variants for associations with XCI ratios within the human data in several slightly different ways to accommodate the cross-tissue sampling structure of the GTEx data (Supplementary Fig. 9A). For the tissue-specific analysis, we rank samples of a given tissue by their XCI ratios and score XCI ratio associations for all variants present within that tissue's samples as previously described using our AUROC metric. We only consider tissues with at least 50 donors. For the donor-specific analysis, we average the tissue XCI ratios of all samples for a given donor and then rank all donors by their average XCI ratio. We also average the allelic-expression ratio of variants present in multiple tissue samples for a given donor and then score XCI ratio associations for all variants as previously described. We additionally perform this experiment using the single tissue sample with the highest sequencing-depth per donor. We down sample the cow sample population to 300 and then score XCI ratio associations for variants as a comparison to the human sample population (314). Sampling is done without replacement 10 times and we compute the average AUROC per variant across the 10 samples (Supplementary Fig. 9).

### Putative mouse XCE-*Xist*-haplotypes

We hierarchically clustered mouse samples by their *Xist* variants considering samples with at least 20 detected *Xist* variants using the ComplexHeatmap[67] R package with the following function options: Heatmap(clustering_distance_columns = function(m dist(m, method = 'binary')), clustering_method_columns = 'ward.D2', column_split = 4). This performs ward.D2 clustering using the Jaccard distance between samples and cuts the column dendrogram using cuttree() into four clusters, which we chose to capture the clear sample groupings present within the data (Supplementary Fig. 10).

### Software

All analysis was performed in R[68] v4.3.3. All plots were generated using ggplot2[69] v3.4.2 functions. The phylogenetic tree in Fig. 2B was generated from TimeTree http://www.timetree.org/.

### Reporting summary

Further information on research design is available in the Nature Portfolio Reporting Summary linked to this article.

## Data availability

The source data for all figure panels can be found at[70] https://github.com/gillislab/cross_mammal_xci/tree/main/R/data_for_plots. Where applicable, exact $p$-values are provided in the source data files. The SRA accession numbers for all non-human mammalian samples processed can be found at https://github.com/gillislab/cross_mammal_

xci/blob/main/R/data_for_plots/all_keep_species_meta.Rdata. Details for accessing the GTEx samples can be found here https://gtexportal.org/home/protectedDataAccess.

## Code availability

All associated code can be found at[70] https://github.com/gillislab/cross_mammal_xci/tree/main/R. Code for generating all figure panels using associated source data can be found at https://github.com/gillislab/cross_mammal_xci/blob/main/R/figure_plots_with_data_code.md. The snakemake pipeline used for processing the non-human mammalian data can be found at https://github.com/gillislab/cross_mammal_xci/tree/main. https://doi.org/10.5281/zenodo.13774726. Details and code for processing the human GTEx samples can be found here[28].

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

## Acknowledgements
J.G., J.M.W., and J.H. were supported by NIH grants R01MH113005. We thank all members of the Gillis lab and particularly John Lee for assisting in some of the initial data downloading.

## Author contributions
J.G. conceived the project. J.M.W. and J.G. designed the experiments and wrote the manuscript. J.M.W. performed the experiments. J.H. and J.M.W performed data management and data processing.

## Competing interests
The authors declare no competing interests.
