## [Transparent Peer Review file · Nature Communications]

Population variability in X-chromosome inactivation across 10 mammalian species

Corresponding Author: Dr Jesse Gillis

Version 0:

Reviewer comments:

Reviewer #1

(Remarks to the Author)

In this study, Werner et al. investigated the factors influencing X chromosome inactivation (XCI) variability, specifically the proportion of inactivated parental alleles, known as the XCI ratio. They analyzed samples from nine mammalian species, including rodents and primates, totaling 9,143 individuals. The study concluded that embryonic stochasticity is a prevalent explanatory model for population XCI variability in mammals, while genetic factors play a comparatively minor role.

Major criticisms:

I appreciate the effort put into the paper by Werner et al., finding it both novel and interesting. However, I find myself slightly unconvinced by the suggestion that XCI patterns are primarily influenced by the stochasticity of early development rather than genotype. In neuroscience, the prevailing view is that the genome significantly shapes early development and contributes to neuropsychiatric diseases. Mutations on the X have been shown to affect XCI ratios.

Additionally, I am seeking further clarification on the relevance of comparing XCI across nine mammalian species when considering XCI patterns in humans. Given the considerable differences observed in XCI among the species analyzed, the comparability might be limited. In essence, it raises questions about the generalizability of mechanisms across species.

I believe what has been presented is very interesting for people in the XCI field but not for a generalized audience.

Minor/Technical.

We don't have detailed knowledge of XCI in all the species studied. Escapees and partial/tissue-specific escapees can affect the analysis. Please elaborate on this point further. Why was the mouse excluded from the analysis? Data from non-inbred mice is publicly available.

This sentence is unclear; please consider rephrasing it: "We focus on the tails of the distributions, 206 as our previous validation using phased data indicated increased uncertainty for folded 207 XCI ratio estimates between 0.5-0.6, which translates to unfolded estimates between 208 0.4-0.6."

There is available data for most of these species; please relate this sentence to existing literature. "Our estimates for the number of epiblast cells present at 210 the time of XCI include 65 (macaca), 31 (rat), 23 (pig), 16 (goat), 15 (horse), 14 (sheep), 211 14 (cow), 13 (human), and 8 (dog) cells, with associated 95% confidence intervals 212 presented in figure 2B. Most importantly, please provide commentary on the human data in function of established XCI literature.

Across all species, X-linked heterozygosity showed a 269 near-zero correlation with the estimated XCI ratio, indicating a lack of association 270 between X-linked genetic variability and XCI ratio variability (Fig. 3B). This is not in line with the current literature. Please provide an exhaustive commentary on this. Indeed, the following paragraph seems to contradict this section.

Reviewer #2

(Remarks to the Author)

In this manuscript, Werner et al. report the degree of mosaicism of X-chromosome inactivation (XCI) in 9 different mammalian species. They address this by exploring allelic information from RNAseq datasets comprising an impressive number of individual samples (9,143). They found that the degree of mosaicism varies considerably among individuals from the 9 different species. Their main conclusion was that embryonic stochasticity as the primary driver of XCI variability across mammalian populations, while genetic factors play a minor role.

Overall, my assessment is positive. Nonetheless, I have a list of important points that the authors should address to strengthen their findings.

- Information on tissues and age of the biological samples used in this study is missing; for instance, XCI skewing increases with aging in blood, but not in other tissues (PMID: 31767861). Therefore, this information is important, specially if blood samples are differently represented across species. This information should be provided.

- The absence of mice from the set of mammalian species utilized in this study is notable. This omission is unlikely due to a scarcity of RNAseq datasets on murine tissues. Instead, it may be attributed to factors such as the significant degree of inbreeding or the known effects of XCE alleles in hybrid mice. Therefore, it would be beneficial for the authors to provide an explanation in the paper regarding their decision to exclude mice from this study.

- In Figure S1, it is observed that the rat (*Rattus norvegicus*) exhibits numerous SNPs with a ratio exceeding 0.75. This is likely attributed to the high degree of inbreeding associated with the utilization of laboratory strains. It would be prudent to acknowledge this observation when initially presenting the figure in the paper.

- In Figure S4, the analysis focused on autosomal allelic imbalances for all samples was limited to two autosomes that are closest in size to the X chromosome. Why was this approach not applied to all chromosomes? Doing so would provide a broader perspective on potential autosomal allelic imbalances across samples. Alternatively, if this was technically unfeasible, clarification on the constraints would be valuable.

- In Figure 2A, the authors derive estimates for the number of epiblast cells present at the onset of XCI by analyzing the unfolded population distributions of XCI ratios across nine mammalian species. How might these estimates be affected by variations in the number of samples and/or informative SNPs? This question gains particular significance in the case of macaques, where both the number of samples and SNPs was the lowest and diverged notably from other species in terms of epiblast cell count. Clarification on these potential influences is essential for a comprehensive understanding of the findings.

- In Figure 4, the authors investigate the potential association between XCI ratios and individual genetic variants on the X chromosome. While the authors rule out the influence of autosomal variants, considering the possibility that autosomal variants, possibly interacting with X-chromosome variants, could impact XCI ratios, would enhance the analysis. Could the inclusion of autosomal variants be considered in this investigation?

- The observation from Figure 4D, wherein many X-linked genes exhibit moderate associations with X-chromosome inactivation (XCI) ratios, particularly strikes attention. Notably, a significant portion, if not the majority, of these genes are recognized escapees in humans (e.g., KDM6A, EIF2S2, XG, MXRA5, MSL3, EIF1AX, TLXNG, DDX3X, IKBKG, RAB9A, P2RY8, IL3RA, ARHGAP6, ARSE, PLXCD1, PLXNB3, ZFX). It is imperative for the authors to acknowledge this fact. Considering that escapees can be expressed from both the active and inactive X chromosomes, the moderate association with XCI ratios is unexpected. What is the authors' interpretation of this observation? Is it linked to a specific expression level associated with a particular variant? Clarification on these points would enrich the discussion.

- In the Discussion section, it would be beneficial for the authors to include a note of caution regarding the need for future experimental validation of their estimation of the number of epiblast cells present at the time of X-chromosome inactivation onset.

Reviewer #3

(Remarks to the Author)

The authors provide some compelling arguments suggesting that random initiation of X chromosome inactivation during embryo development almost completely explain the variability of XCI in adult tissues across 9 mammalian species, and that genetic factors play only a very minor role. This is an important question in the field of XCI (and by extension epigenetic in general). They perform sound statistical analysis and their data are clear and easily interpretable.

The fact that random XCI initiation in embryo at a certain cell stage is sufficient to explain the variability in XCI is particularly interesting. It would be worthwhile to include a table comparing the model's prediction to experimental data that have observed the embryonic stage at which XCI is initiated in the different species (when data are available).

While covering 9 mammalian species, their analysis did not include mice, which is a very common model for the study of XCI and one where a large amount of data are available. While the results might be very close to those of the rats (included in the study), it would be beneficial for the field to have mice data included here.

While the statistical analysis are generally very clear, the notions of "folded" and "unfolded" distribution might be hard to understand for most scientists in the field of XCI who are not highly versed in statistics. The authors could add a few sentences explaining in layman's terms what this represents and use more descriptive terms when describing their figures.

For example, the sentence “we then generate population-level distributions by unfolding the 164 distribution of folded XCI ratio sample estimates per species” line 164-5 might be difficult to understand by some readers.

In figure 1A, the authors fit a distribution to “the tail of the empirical distribution”. While the “tail” are indicated in colors, the author should clearly define the value of the thresholds for the tail (line 208 “XCI ratio estimates between 0.5-0.6, which translates to unfolded estimates between 208 0.4-0.6.

” are 0.4-0.6 the thresholds used? If so this could be mentioned more clearly).

Line 206 The authors also mention “We focus on the tails of the distributions, as our previous validation using phased data indicated” the publication for the “previous validation” should be cited here.

In supplementary figure 4, it is unclear what is the value of the x axis “chrX estimated skew”. The authors should clarify what is exactly the value shown here.

Line 220 the author states “The reported X-linked variability in macaca is in excess to the reported autosomal allelic variability” It is a bit unclear what the authors mean by that and what data support that claim. Are the author referring to the fact that in the 2 plots on top right in Sup Fig 4A, the chrX estimated skew is very slightly higher than the autosomal imbalance in some samples (subsets of dots in the plot)?

In figure 2A , most distributions show a dip around 0.5. Do the authors have an explanation for that?

Fig2 B second panel : to be easier to interpret, the X axis should be labelled in non log scale (absolute number of cells) and the ticks can be log scaled.

Version 1:

Reviewer comments:

Reviewer #2

(Remarks to the Author)

The authors replied positively to my original concerns.

Reviewer #3

(Remarks to the Author)

I would like to thank the authors for addressing all comments and request for clarifications in a clear and detailed manner. I have no further comments and you support publication of the manuscript.

RESPONSE TO REVIEWERS' COMMENTS

Reviewer #1 (Remarks to the Author)

In this study, Werner et al. investigated the factors influencing X chromosome inactivation (XCI) variability, specifically the proportion of inactivated parental alleles, known as the XCI ratio. They analyzed samples from nine mammalian species, including rodents and primates, totaling 9,143 individuals. The study concluded that embryonic stochasticity is a prevalent explanatory model for population XCI variability in mammals, while genetic factors play a comparatively minor role.

Major criticisms:

I appreciate the effort put into the paper by Werner et al., finding it both novel and interesting. However, I find myself slightly unconvinced by the suggestion that XCI patterns are primarily influenced by the stochasticity of early development rather than genotype. In neuroscience, the prevailing view is that the genome significantly shapes early development and contributes to neuropsychiatric diseases. Mutations on the X have been shown to affect XCI ratios.

We thank the reviewer for this comment and have edited the language of our conclusions in this work to better reflect the existing known roles of genetics on XCI ratios. It is not our intent to disregard this prior work and we have included citation to numerous examples that establish genetic associations with XCI ratios. We think there is an important distinction to draw in our population scale analysis from samples based on disease. We observe that models of stochasticity explain the observed population-level XCI ratio variability across species exceptionally well, while we consistently fail to find genetic associations. Note that because this is only a statement about allelic ratios, it has almost no direct bearing on the degree to which the genome shapes development (unless the two Xs are very different). In the population at large, the purifying selection the X is under in males will tend to diminish the functional importance of allele choice, except in the case of disease. We do not conclude this means there are no genetic associations with XCI, rather we conclude that embryonic stochasticity offers a better explanation for population variability compared to genetics outside of a disease context. In the edited text, we also include the point that our approach only samples genetic variants present within transcribed RNA molecules and we are likely missing a large amount of non-transcribed regulatory variants that may have a larger impact on XCI ratios. While we do not find pervasive genetic associations with XCI ratios in the data we collected, we are limited on the type of genetic variability we can assess and we make that clearer in the current text.

Lines 151 – 153: “Additionally, it is important to note our approach detects SNPs present only within RNA molecules, so we will miss variants in non-transcribed proximal regulatory elements, such as the well-described XCE-interval in mice³³.”

Lines 395 – 400: “We showed that models of embryonic stochasticity fit the XCI data exceptionally well and estimated epiblast cell counts at the time of XCI across species. We also searched for genetic factors influencing XCI ratios and found a pervasive lack of strong genetic associations with XCI ratios, indicating that population XCI variability is better explained by the inherent stochastic nature of XCI rather than through genetic mechanisms.”

Lines 469 – 488: “An important caveat to our analyses of genetic influences on XCI ratios is that we are limited to assessing variants present within RNA molecules, which are necessary for quantifying allele-specific expression. Consequently, we likely miss many non-transcribed regulatory variants that may significantly influence XCI ratios. The XCE-interval in mice is one example, a proximal regulatory element to *Xist* known to influence XCI ratios in heterozygous lab crosses³³. We identified putative XCE-haplotypes in the DO mice population using *Xist* variants as proxies and showed XCI ratios are highly variable among samples sharing a haplotype, indicating other factors outside of *Xist* genotypes/putative XCE-haplotypes influence XCI ratios in DO mice. While this suggests the effects of XCE-haplotypes in DO mice are minor and more pronounced in inbred lab crosses, we cannot exclude the possibility of similar genetic influences on XCI ratios in non-transcribed regions among our sampled species. Our approach trades comprehensive genetic screening for scalability, enabling the assessment of XCI ratios across thousands of samples from different species. While we demonstrate models of embryonic stochasticity explain observed population XCI ratio variability far better than genetic associations, we are limited in the type of genetic variability we can assess. Importantly, we do not discount the well-documented role of genetics in XCI ratios for rare and exceptional cases (e.g., disease); instead, we advance embryonic stochasticity as the parsimonious explanation of XCI variability in normal populations.”

Additionally, I am seeking further clarification on the relevance of comparing XCI across nine mammalian species when considering XCI patterns in humans. Given the considerable differences observed in XCI among the species analyzed, the comparability might be limited. In essence, it raises questions about the generalizability of mechanisms across species.

We agree that comparability might be limited, which is why we think the similarity of distributions in many cases (and the exceptions) is quite interesting. This observation is also broadly consistent with a view of embryonic stochasticity as a driver of population variance, since it is shared if the timing of X-inactivation is shared. Similarly, while exact mechanisms (and SNP associations) might differ, the overall prevalence across species of such mechanisms is a generally useful assay into the importance of genetic factors in XCI variance, as compared to stochasticity or genetic incompatibility.

I believe what has been presented is very interesting for people in the XCI field but not for a generalized audience.

We appreciate the positive side of this comment and while we don't wish to push back too hard on the negative, we do think the implications for X-linked phenotypes (including disease) are not trivial and because we use publicly available data, each of our species results has their own community of interest, in addition to any cross-species comparisons.

Minor/Technical.

We don't have detailed knowledge of XCI in all the species studied. Escapees and partial/tissue-specific escapees can affect the analysis. Please elaborate on this point further. Why was the mouse excluded from the analysis? Data from non-inbred mice is publicly available.

We agree that genes which escape XCI can impact our analysis and we employ an approach to identify and exclude regions of the X-chromosome that exhibit signal of XCI escape (Supp. Fig. 2). Our approach to estimate XCI ratios also aggregates information across many genes per sample, averaging out the impact of potential escape, which we did explore extensively in our previous work (PMID: 35914524)

We initially excluded data from inbred mice strains for the reasons of high levels of inbreeding and XCE-haplotype effects, but we have now included 388 mouse samples from 2 studies that sampled bulk RNA-seq from Diversity Outbred mice. Analysis for the mouse samples is now included in all of the main figures. Diversity Outbred (DO) mice control for the effects of inbreeding, which we observe via low levels of reference-biased SNPs and a high degree of X-linked heterozygosity as compared to the other non-human species (Supp. Fig. 1, Fig. 3A), while the effects of XCE-haplotypes are less clear. To our knowledge, the role of XCE-haplotypes in influencing XCI ratios has only been studied within inbred heterozygous crosses and not within DO genetic backgrounds. We attempt to identify putative XCE-*Xist*-haplotypes within the DO mice using *Xist* variants as proxies for XCE-haplotypes (Supp. Fig. 10). We are only able to identify transcribed variants such as within *Xist* and we assume XCE and *Xist* variants are likely linked due to their proximity. In general, we observe that XCI ratios of samples that share putative XCE-*Xist*-haplotypes are highly variable and there is no clear strong bias in XCI ratios dependent on the XCE-*Xist*-haplotypes, all of which suggests XCI ratios in DO mice have non-genetic contributions outside of XCE-effects. We have included the following text in the Results section as well as an additional Supplemental Figure 10 for this data.

Lines 366 – 389:

“Putative mouse XCE-*Xist*-haplotypes exhibit highly variable XCI ratios

One of the most well-documented instances of a genetic association with XCI ratios is the XCE-haplotypes in laboratory mouse strains, where a preferential ordering of allelic

inactivation exists across haplotypes³³. The DO mice we utilize are expected to be genetically diverse combinations of various lab strains and it is highly likely a mix of XCE-haplotypes are present within this population and may have an impact on XCI ratios. Such a haplotype-specific effect would be missed in our previous AUROC variant-specific analysis of XCI ratios. Since we only sample variants present within RNA molecules and the XCE-interval is a proximal non-transcribed regulatory element of *Xist*³³, we reason variants present within *Xist* are likely linked to XCE-haplotypes and may be informative for identifying putative XCE-*Xist*-haplotypes. We identify 4 putative XCE-*Xist*-haplotypes as determined by groups of samples with shared *Xist* variants (Supp. Fig. 10). As a general observation across haplotypes and the two studies we sample from, XCI ratios of samples with the same haplotype are highly variable (Supp. Fig. 10), suggesting XCI ratios are not definitively determined by *Xist* genotypes within the DO mice population. The haplotype with the seemingly largest effect, evidenced by 2 samples with highly skewed XCI ratios (0.799 and 0.782) in the one study that collected striatal tissue, conversely exhibits XCI ratios ranging from balanced to moderately skewed in the second study, which collected pancreatic tissue (Supp. Fig. 10). While this may be indicative of a tissue-specific effect of a particular XCE-*Xist*-haplotype, far greater sample sizes with higher genetic resolution to confirm haplotypes are needed for validation. In general, the variability in XCI ratios within putative XCE-*Xist*-haplotypes suggests non-genetic contributions to XCI variability of DO mice.”

This sentence is unclear; please consider rephrasing it: “We focus on the tails of the distributions, 206 as our previous validation using phased data indicated increased uncertainty for folded 207 XCI ratio estimates between 0.5-0.6, which translates to unfolded estimates between 208 0.4-0.6.”

We agree the text can be clearer and have made the following edits:

Lines 221 – 228: “We focus on the tails of the distributions for our model fitting (colored in portions of the distributions, unfolded estimates ≤ 0.40 and ≥ 0.60 , Fig. 2A), for two reasons. Our analysis of autosomal allelic ratios (Supp. Fig. 5) highlights that samples with no expected allelic imbalance produce folded skew estimates that vary between 0.5 and 0.6 and our previous work²⁸ using phased data indicated model misspecification around the point of folding (0.50). Fitting to the tails of the empirical distribution is therefore a more accurate representation of variability specific to XCI.”

There is available data for most of these species; please relate this sentence to existing literature. “Our estimates for the number of epiblast cells present at 210 the time of XCI include 65 (macaca), 31 (rat), 23 (pig), 16 (goat), 15 (horse), 14 (sheep), 211 14 (cow), 13 (human), and 8 (dog) cells, with associated 95% confidence intervals 212 presented in figure 2B. Most importantly, please provide commentary on the human data in function of established XCI literature.

We thank the reviewer for this comment and agree we can provide better context for our results by incorporating prior literature. We do respectfully disagree with the reviewer that data exists on the number of cells present at the time of XCI across the species we sample, due to the fact XCI timing is still debated across species, even for the arguably most studied species (mice, macaca, and humans) and cell counts are a rarely reported quantification. While the hallmarks of inactivation in mice are clear (namely mono-allelic expression of XIST) and the timing of mouse XCI is understood to occur shortly after epiblast specification, time course data in macaca and human show a more continuous shift from biallelic to mono-allelic expression that occurs over several days (PMID: 34793202, PMID: 28883481, PMID: 27062923), obscuring the exact timing of XCI. Indeed, initial biallelic expression of XIST appears to be more common across species than not (PMID: 23578369), highlighting that hallmarks of inactivation in mice are not easily applicable across species. To our knowledge, detailed time course data quantifying X-linked allelic expression is not widely available across the species we sample, and even when available, different conclusions have been drawn on the timing of XCI across studies, using human data as an example (PMID: 28883481, PMID: 27062923). Specifically for our epiblast cell counts, these cell numbers represent cell population sizes at the time of XCI and are not reflective of the exact timing of XCI across species. For example, more cells could be present at the time of XCI in macaca compared to human due to delayed XCI in macaca allowing more cell divisions to occur or more cells are initially specified for the epiblast. Variability in cell counts could be a result in variability in XCI timing or variability in cell counts during lineage specification events. We have edited the corresponding results and discussion text to better reflect these points.

Results lines 260 – 276: “Modeling XCI ratio variability across numerous species allows comparisons in light of evolution for determining generalizable or species-specific characteristics in XCI. Broadly, we demonstrate XCI ratios are variable in each species we assess, revealing variability in XCI ratios itself as a conserved characteristic of XCI. The exact variance in XCI ratios varies across the species, with differences in the timing of XCI and/or differences in cell counts for embryonic/extra-embryonic lineage specification as two putative explanations. We compare our estimated cell counts to the evolutionary relationships among the species we assess (Fig. 2B), suggesting that variability in these early embryonic events can be recent evolutionary adaptations. This is highlighted by the large differences in cell counts between macaca and humans, as well as between rats and mice. When viewed through the lens of cell divisions (\log_2 of the estimated cell counts, Fig. 2B), the differences in XCI ratio variability among the species can be explained by differences in a range of only 3 cell divisions, a narrow developmental window. This demonstrates even slight changes in the timing of XCI or cell counts for embryonic/extra-embryonic lineage specification across mammalian species can produce large differences in population XCI ratio variability, as explained through the inherent stochasticity of XCI.”

Discussion lines 451 – 467: “Regarding the timing of XCI and our epiblast cell count estimates, exploring known temporal variability in XCI across species provides additional context. In mice, random XCI occurs within the epiblast soon after its

specification and is readily identified by monoallelic expression of XIST⁵⁵. In macacas and humans, inactivation appears more continuous; data show progressive chromosome-wide silencing over several days, with a shift from biallelic to monoallelic XIST expression^{56–58}. This lengthy continuous inactivation obscures the exact timing of XCI in these species and highlights that XCI hallmarks in mice, namely rapid monoallelic XIST expression, are not readily applicable to other species; indeed, many species initially exhibit biallelic XIST expression⁵⁹. In context, our epiblast cell counts estimate the number of cells involved in XCI, not the exact timing. For instance, our macaca and human cell counts indicate approximately four times as many cells are present within the epiblast at the time of XCI in macacas compared to humans. This difference could result from delayed XCI or a greater number of cells fated for the epiblast in macacas compared to humans. In general, our cell count estimates reflect population sizes at the time of XCI and we attribute variability in cell counts as most likely due to differences in XCI timing or lineage specification dynamics across species.”

Across all species, X-linked heterozygosity showed a 269 near-zero correlation with the estimated XCI ratio, indicating a lack of association 270 between X-linked genetic variability and XCI ratio variability (Fig. 3B). This is not in line with the current literature. Please provide an exhaustive commentary on this. Indeed, the following paragraph seems to contradict this section.

To our knowledge, this is the first assessment of broad X-chromosome heterozygosity as it relates to XCI ratios. The vast majority of known genetic associations with XCI ratios are derived from single-gene/variant studies and often in disease settings, which we cite in the Introduction and Discussion sections. There is a prevalent hypothesis that general heterozygosity on the X can drive XCI ratios via selection due to observations of selective effects in disease settings, but to our knowledge this has never been tested outside of a disease context, especially across species. We have edited the relevant Discussion text to better highlight this distinction, included below. Note that this also explains why “Low frequency variants exhibit moderate associations with XCI ratios” is compatible with our population observations. We first assess broad X-chromosome heterozygosity and then we assess individual variants for XCI ratio associations, where we think the two can be mutually exclusive. Indeed, this puts our results on a sort of spectrum with published disease results where most XCI is stochastic unless something (rarely) is genetically associated, with the most extreme such cases being disease.

Discussion lines 407 – 426: “Although genetic influences on XCI, particularly variants affecting XIST¹⁵ or disease-associated variants^{34–37}, have been identified, they do not constitute a general mechanism that can fully account for observed population-level XCI variability across species. In particular, allelic selection via genetic variability across the X-alleles has been put forth as an explanatory mechanism for XCI ratio variability^{14,16}, but is almost exclusively studied in a disease context and typically associated with extreme XCI ratios^{14,16,17,34–38}, which is conflicting with the continuous population variability we report across species. Our measures of X-linked heterozygosity have

near-zero association with XCI ratios in all 10 mammals we assessed, a strong indication that genetic variability on the X-chromosome has little influence on XCI ratios outside of disease. This is supported by the observation of depleted X-chromosome genetic variability via strong rates of purifying selection^{50–53}, rendering both parental alleles as largely equivalent at population scales. Our approach for extracting heterozygous variants from RNA-seq data²⁸, while providing a sample of genetic variability, is still able to assess hundreds of X-linked genes and chromosome-wide heterozygosity per species for associations with XCI and culminated in only weak evidence of limited genetic influence on XCI ratios. In contrast, we demonstrated models of embryonic stochasticity can explain population XCI variability with exceedingly small amounts of error consistently across mammalian species, providing a much more general explanation for population XCI variability.”

Reviewer #2 (Remarks to the Author)

In this manuscript, Werner et al. report the degree of mosaicism of X-chromosome inactivation (XCI) in 9 different mammalian species. They address this by exploring allelic information from RNAseq datasets comprising an impressive number of individual samples (9,143). They found that the degree of mosaicism varies considerably among individuals from the 9 different species. Their main conclusion was that embryonic stochasticity as the primary driver of XCI variability across mammalian populations, while genetic factors play a minor role. Overall, my assessment is positive. Nonetheless, I have a list of important points that the authors should address to strengthen their findings.

- Information on tissues and age of the biological samples used in this study is missing; for instance, XCI skewing increases with aging in blood, but not in other tissues (PMID: 31767861). Therefore, this information is important, specially if blood samples are differently represented across species. This information should be provided.

We agree with the reviewer that tissue and age annotations would be highly informative in this study, but such information is regrettably not available. In order to assess thousands of samples, we source data from SRA where metadata is incomplete more often than not. Out of the 13,928 samples we assessed excluding the human and mouse samples, only 25 had tissue annotations in SRA (Macaca:4, Goat:3, Sheep:9, Pig:9) and age is entirely absent. We sample data over 562 individual studies, making it infeasible to collect sample metadata manually. We've added the following text to the discussion section to acknowledge this limitation of our study.

Lines 442-444: “In general, we cannot account for tissue-specific or age-related effects in this dataset as these sample annotations are almost universally absent for the SRA sourced data.”

We did investigate relationships between age and XCI ratios in the GTEx data where age annotations are available and only recovered a mild age association for a few tissues. Here, we quantify whether XCI ratios can predict GTEx donors at least 60 years or older using the AUROC statistic. A high AUROC means ranking XCI ratios of a tissue places the older samples at the top of the ranked list. None of the AUROCs reach statistical significance and the tissues with the highest AUROCs (~0.6, top 3 shown below) show only a minor trend between age and XCI ratios. We also do not report a strong association between age and blood samples, though this may be influenced by the coarse binning of the age annotations made available from GTEx. We expect these results might be different if more extreme ages at finer resolution were sampled but our results suggest the effect is modest over sampled ages.

- The absence of mice from the set of mammalian species utilized in this study is notable. This omission is unlikely due to a scarcity of RNAseq datasets on murine tissues. Instead, it may be attributed to factors such as the significant degree of inbreeding or the known effects of XCE alleles in hybrid mice. Therefore, it would be beneficial for the authors to provide an explanation in the paper regarding their decision to exclude mice from this study.

We did indeed initially exclude data from inbred mice strains for the exact reasons of high levels of inbreeding and XCE-haplotype effects, but we have now included 388 mouse samples from 2 studies that sampled bulk RNA-seq from Diversity Outbred mice. Diversity Outbred (DO) mice control for the effects of inbreeding, which we observe via low levels of reference-biased SNPs and a high degree of X-linked heterozygosity as compared to the other non-human species (Supp. Fig. 1, Fig. 3A), while the effects of XCE-haplotypes are less clear. To our knowledge, the role of XCE-haplotypes in influencing XCI ratios has only been studied within inbred heterozygous crosses and not within DO genetic backgrounds. We attempt to identify putative XCE-

Xist-haplotypes within the DO mice using *Xist* variants as proxies for XCE-haplotypes (Supp. Fig. 10). We are only able to identify transcribed variants such as within *Xist* and we assume XCE and *Xist* variants are likely linked due to their proximity. In general, we observe that XCI ratios of samples that share putative XCE-*Xist*-haplotypes are highly variable and there is no clear strong bias in XCI ratios dependent on the XCE-*Xist*-haplotypes, all of which suggests XCI ratios in DO mice have non-genetic contributions outside of XCE-effects. We have included the following text in the Results section as well as an additional Supplemental Figure 10 for this data.

Lines 366 – 389:

“Putative mouse XCE-*Xist*-haplotypes exhibit highly variable XCI ratios

One of the most well-documented instances of a genetic association with XCI ratios is the XCE-haplotypes in laboratory mouse strains, where a preferential ordering of allelic inactivation exists across haplotypes³³. The DO mice we utilize are expected to be genetically diverse combinations of various lab strains and it is highly likely a mix of XCE-haplotypes are present within this population and may have an impact on XCI ratios. Such a haplotype-specific effect would be missed in our previous AUROC variant-specific analysis of XCI ratios. Since we only sample variants present within RNA molecules and the XCE-interval is a proximal non-transcribed regulatory element of *Xist*³³, we reason variants present within *Xist* are likely linked to XCE-haplotypes and may be informative for identifying putative XCE-*Xist*-haplotypes. We identify 4 putative XCE-*Xist*-haplotypes as determined by groups of samples with shared *Xist* variants (Supp. Fig. 10). As a general observation across haplotypes and the two studies we sample from, XCI ratios of samples with the same haplotype are highly variable (Supp. Fig. 10), suggesting XCI ratios are not definitively determined by *Xist* genotypes within the DO mice population. The haplotype with the seemingly largest effect, evidenced by 2 samples with highly skewed XCI ratios (0.799 and 0.782) in the one study that collected striatal tissue, conversely exhibits XCI ratios ranging from balanced to moderately skewed in the second study, which collected pancreatic tissue (Supp. Fig. 10). While this may be indicative of a tissue-specific effect of a particular XCE-*Xist*-haplotype, far greater sample sizes with higher genetic resolution to confirm haplotypes are needed for validation. In general, the variability in XCI ratios within putative XCE-*Xist*-haplotypes suggests non-genetic contributions to XCI variability of DO mice.”

- In Figure S1, it is observed that the rat (*Rattus norvegicus*) exhibits numerous SNPs with a ratio exceeding 0.75. This is likely attributed to the high degree of inbreeding associated with the utilization of laboratory strains. It would be prudent to acknowledge this observation when initially presenting the figure in the paper.

We agree and have edited the corresponding Results text accordingly, included below.

Lines 142 – 150: “We also incorporate specific steps to address confounding factors that can impact measured X-linked allelic expression, namely excluding SNPs with persistent reference bias across samples and chromosomal bins that exhibit probable

escape from XCI^{40,41} (Supp. Figs. 1-2, see methods). Of note, the rat population exhibits a large collection of reference biased SNPs when compared to the other species, likely due to the highly inbred nature of laboratory rat strains. We circumvent this expected issue in the mouse population by leveraging two studies^{42,43} that sampled Diversity Outbred (DO)⁴⁴ mice, evidenced by the lack of reference-biased SNPs in the mouse population compared to the other species.”

- In Figure S4, the analysis focused on autosomal allelic imbalances for all samples was limited to two autosomes that are closest in size to the X chromosome. Why was this approach not applied to all chromosomes? Doing so would provide a broader perspective on potential autosomal allelic imbalances across samples. Alternatively, if this was technically unfeasible, clarification on the constraints would be valuable.

We limited autosomal processing due to data storage and time constraints. Just the X-chromosome and the current 2 autosomes per species accounts for 183 terabytes of data storage and took several months of compute time once the processing pipeline was finalized. Extending this analysis across all autosomes for all species would entail prohibitive storage requirements and a considerable amount of time.

- In Figure 2A, the authors derive estimates for the number of epiblast cells present at the onset of XCI by analyzing the unfolded population distributions of XCI ratios across nine mammalian species. How might these estimates be affected by variations in the number of samples and/or informative SNPs? This question gains particular significance in the case of macaques, where both the number of samples and SNPs was the lowest and diverged notably from other species in terms of epiblast cell count. Clarification on these potential influences is essential for a comprehensive understanding of the findings.

We now include comparisons between our estimated cell counts and the mean number of well-powered SNPs as well as total sample size per species in Supplemental Figure 6, included below. We note that dog, rat, and macaca, which all have a similar mean number of SNPs per sample, vary substantially in their estimated cell numbers. We also include a new analysis where we down-sample species to the Macaca sample size and estimate their cell counts, producing virtually identical cell number estimates. Overall, these new comparisons highlight our cell number estimates are not driven by the number of detected SNPs or population sample size across the species. We have edited the corresponding Results section as follows:

Lines 234 – 244: “Importantly, species with similar numbers of detected SNPs per sample and total sample size exhibit variable cell number estimates, indicating it is unlikely our estimates are driven by technical effects across the species (Supp. Fig. 6). We additionally down sample all species to the smallest sample size present (130

macaca samples) and achieve virtually identical cell number estimates, demonstrating variation in cell number estimates across species is not driven by sample size differences (Supp. Fig. 6). The error between the empirical XCI ratio distributions and the normal fitted distributions is strikingly small, with a mean of 0.00588 sum-squared error (± 0.00965 SD) across the species (Supp. Fig. 6). This shows models of embryonic stochasticity can explain observed XCI ratio variability in adult populations exceptionally well.”

Supplemental Figure 6B-D

- In Figure 4, the authors investigate the potential association between XCI ratios and individual genetic variants on the X chromosome. While the authors rule out the influence of autosomal variants, considering the possibility that autosomal variants, possibly interacting with X-chromosome variants, could impact XCI ratios, would enhance the analysis. Could the inclusion of autosomal variants be considered in this investigation?

While we agree that investigating potential interactions between the autosomes and X-chromosome is of general interest, the aforementioned technical constraints of including autosomal data limit our ability to perform such an analysis.

- The observation from Figure 4D, wherein many X-linked genes exhibit moderate associations with X-chromosome inactivation (XCI) ratios, particularly strikes attention. Notably, a significant portion, if not the majority, of these genes are recognized escapees in humans (e.g., KDM6A, EIF2S2, XG, MXRA5, MSL3, EIF1AX, TLXNG, DDX3X, IKBKG, RAB9A, P2RY8, IL3RA, ARHGAP6, ARSE, PLXCD1, PLXNB3, ZFX). It is imperative for the authors to acknowledge this fact. Considering that escapees can be expressed from both the active and inactive X chromosomes, the moderate association with XCI ratios is unexpected. What is the authors' interpretation of this observation? Is it linked to a specific expression level associated with a particular variant? Clarification on these points would enrich the discussion.

We thank the reviewer for highlighting this point and we agree that it is not clear why a variant in an escapee would have an association with XCI ratios. To confirm our AUROC statistic was in fact identifying variants with XCI ratio associations within these XCI escapees, we compare the allelic-expression ratios of these variants to the estimated XCI ratios of the tissue samples where the variants are present. These comparisons show an interesting trend where the allelic-expression of the variant does show signal of escape from XCI, namely allelic-expression close to 0.50, while the variant is also present in samples with clearly increased XCI ratios, Supplemental Figure 9C included below. This confirms our AUROC statistic is identifying variants associated with XCI ratios and confirms these variants in escapees seem to also escape XCI. Since the mechanism through which a biallelically expressed variant may be associated with XCI ratios is unclear, we think these variants may be present on haplotypes that have an association with XCI ratios instead, rather than the specific gene/variant. We have edited the Results text as follows:

Lines 350 – 364: “Several genes with moderate AUROCs have prior evidence for escaping XCI in humans⁴⁹, bringing into question their associations with extreme XCI ratios in our analysis. To explore further, we compare the estimated XCI ratios of samples to the allelic ratios of all detected variants for genes with at least one variant significantly associated with XCI ratios. We report several examples across species where the allelic expression of individual variants from these putative escape genes does in fact exhibit the expected balanced biallelic expression of escape from XCI while also being enriched in samples with increased XCI ratios (Supp. Fig. 9). A gene that escapes XCI will be biallelically expressed; this suggests the variant-specific association we detect within these XCI escapees likely reflects a haplotype-effect, where the variant is linked to a haplotype influencing XCI ratios, rather than an effect from the gene/variant itself. Further analysis with phased data to assess potential haplotype effects may help identify genetic associations with XCI ratios. Overall, our assessments of chromosome-wide genetic variability and individual variants do not reveal genetic associations robust enough to explain population XCI ratio variability across all 10 mammalian species.”

We would also like to note an update to our results regarding the human variant-XCI ratio associations. In our previous results, a variant that was only present in a single GTEx donor but detected in multiple tissues for that donor was assessed as a variant present across independent samples. The variants we previously identified as significantly associated with XCI ratios in humans were almost all present in individual donors whose tissues were all skewed for XCI, producing statistics that did not reflect the true sample size of the human population. We have updated our approach for identifying XCI associated variants in the human population to account for this, namely performing a tissue-specific and donor-specific experiment as well as using the sample per donor with the highest sequencing depth. Across these three different approaches, we no longer detect any significant variant-XCI ratio associations in the human population, which we attribute to GTEx’s small sample size (314 individuals). We down sample the cow data to 300 as a comparison and show all the variants we called as

significantly associated with XCI in cow are not detected in the down sampled simulations, in agreement with our initial observation that XCI associated variants have low population frequencies. We have updated the corresponding Results and Methods text and include the new Supplementary Figure 9 below:

Results lines 337 – 348: “Notably, we observe no statistically significant variant-XCI ratio associations in the GTEx human population when performing either a tissue-specific or donor-specific analysis, as well as only considering the sample per donor with the highest sequencing depth (Supp. Fig. 9). While the GTEx dataset is comprised of thousands of tissue samples, only 314 female individuals are present in our final dataset. We test the effect of a small population size by down sampling the cow data to 300 samples and scoring variant-XCI ratio associations (Supp. Fig. 9). All of the cow variants that we originally identified as significantly associated with XCI ratios are no longer detected in the down sampled data, in line with the observation that variants with associations to XCI ratios occur at low frequencies within mammalian populations. Increased population sampling is likely required to identify further genetic associations with XCI ratios.”

Methods lines 689 – 703: “We assess variants for associations with XCI ratios within the human data in several slightly different ways to accommodate the cross-tissue sampling structure of the GTEx data (Supp. Fig. 9A). For the tissue-specific analysis, we rank samples of a given tissue by their XCI ratios and score XCI ratio associations for all variants present within that tissue’s samples as previously described using our AUROC metric. We only consider tissues with at least 50 donors. For the donor-specific analysis, we average the tissue XCI ratios of all samples for a given donor and then rank all donors by their average XCI ratio. We also average the allelic-expression ratio of variants present in multiple tissue samples for a given donor and then score XCI ratio associations for all variants as previously described. We additionally perform this experiment using the single tissue sample with the highest sequencing-depth per donor. We down sample the cow sample population to 300 and then score XCI ratio associations for variants as a comparison to the human sample population (314). Sampling is done without replacement 10 times and we compute the average AUROC per variant across the 10 samples (Supp. Fig. 9).”
Supplemental Figure 9C

Supplemental Figure 9: Human variant XCI ratio associations and example cross-variant allelic expression ratios within individual genes

A Scatter plots comparing the AUROC (x-axis) of a variant to its FDR-corrected p-value (y-axis) for human variants only. P-values are reported in negative log-10 scale. Variants with an FDR-corrected p-value ≤ 0.05 are highlighted in red. The plots report the statistics for human variants assessed in a tissue-specific manner (leftmost), donor-specific manner (center), and when using the sample per donor with the highest sequencing depth (rightmost). **B** Scatter plots comparing the original AUROC (x-axis) to the mean AUROC using down sampled data (y-axis) for variants within the 4 cow genes that reported significant XCI ratio associations (AUROC ≥ 0.75 , red arrows). Red lines indicate the 0.75 threshold, all of the variants with original AUROCs ≥ 0.75 are not detected in the down sampled data. **C** Boxplots comparing the estimated XCI ratios of

tissues (red, y-axis) and the allelic-ratios of individual variants (blue, y-axis) within those tissues across all variants detected within a given gene (x-axis). We provide examples of genes for several species that contain at least 1 variant we identify as significantly associated with XCI ratios (AUROC \geq 0.75, red asterisks). Escape annotations are from Supp. Table 1 here⁴⁹.

- In the Discussion section, it would be beneficial for the authors to include a note of caution regarding the need for future experimental validation of their estimation of the number of epiblast cells present at the time of X-chromosome inactivation onset.

We agree with the reviewer and have added the following text in the Discussion section:

Lines 444 – 449: “These factors indicate that our epiblast cell count estimates are lower bound estimates for the number of cells needed to produce the observed XCI ratio variability as purely derived from embryonic stochasticity. Our statistical modeling approach here would be greatly complemented by future experimental validation of the timing and cell counts present during XCI across species.”

Reviewer #3 (Remarks to the Author):

The authors provide some compelling arguments suggesting that random initiation of X chromosome inactivation during embryo development almost completely explain the variability of XCI in adult tissues across 9 mammalian species, and that genetic factors play only a very minor role. This is an important question in the field of XCI (and by extension epigenetic in general). They perform sound statistical analysis and their data are clear and easily interpretable.

The fact that random XCI initiation in embryo at a certain cell stage is sufficient to explain the variability in XCI is particularly interesting. It would be worthwhile to include a table comparing the model's prediction to experimental data that have observed the embryonic stage at which XCI is initiated in the different species (when data are available).

We agree with the reviewer that comparing our results to the known timing of XCI across species is important and we have included a new Discussion paragraph to address this point. We would like to clarify that our cell count estimates are not estimates of the cell-stage at which XCI occurs, but rather an estimate of the number of cells that are involved in XCI. Specifically, the number of cells fated for the epiblast at the time of XCI, since we only sample variability from the embryonic lineage via sampling adult tissues. Additionally, the exact timing of XCI across species is still an area of active research and is generally unclear for most species, limiting the data available for the requested comparison. The following is a new Discussion paragraph detailing these points:

Discussion lines 451 – 467: “Regarding the timing of XCI and our epiblast cell count estimates, exploring known temporal variability in XCI across species provides additional context. In mice, random XCI occurs within the epiblast soon after its specification and is readily identified by monoallelic expression of XIST⁵⁵. In macacas and humans, inactivation appears more continuous; data show progressive chromosome-wide silencing over several days, with a shift from biallelic to monoallelic XIST expression^{56–58}. This lengthy continuous inactivation obscures the exact timing of XCI in these species and highlights that XCI hallmarks in mice, namely rapid monoallelic XIST expression, are not readily applicable to other species; indeed, many species initially exhibit biallelic XIST expression⁵⁹. In context, our epiblast cell counts estimate the number of cells involved in XCI, not the exact timing. For instance, our macaca and human cell counts indicate approximately four times as many cells are present within the epiblast at the time of XCI in macacas compared to humans. This difference could result from delayed XCI or a greater number of cells fated for the epiblast in macacas compared to humans. In general, our cell count estimates reflect population sizes at the time of XCI and we attribute variability in cell counts as most likely due to differences in XCI timing or lineage specification dynamics across species.”

While covering 9 mammalian species, their analysis did not include mice, which is a very common model for the study of XCI and one where a large amount of data are available. While the results might be very close to those of the rats (included in the study), it would be beneficial for the field to have mice data included here.

We initially excluded data from inbred mice strains for the reasons of high levels of inbreeding and XCE-haplotype effects, but we have now included 388 mouse samples from 2 studies that sampled bulk RNA-seq from Diversity Outbred mice. Analysis for the mouse samples is now included in all of the main figures. Diversity Outbred (DO) mice control for the effects of inbreeding, which we observe via low levels of reference-biased SNPs and a high degree of X-linked heterozygosity as compared to the other non-human species (Supp. Fig. 1, Fig. 3A), while the effects of XCE-haplotypes are less clear. To our knowledge, the role of XCE-haplotypes in influencing XCI ratios has only been studied within inbred heterozygous crosses and not within DO genetic backgrounds. We attempt to identify putative XCE-*Xist*-haplotypes within the DO mice using *Xist* variants as proxies for XCE-haplotypes (Supp. Fig. 10). We are only able to identify transcribed variants such as within *Xist* and we assume XCE and *Xist* variants are likely linked due to their proximity. In general, we observe that XCI ratios of samples that share putative XCE-*Xist*-haplotypes are highly variable and there is no clear strong bias in XCI ratios dependent on the XCE-*Xist*-haplotypes, all of which suggests XCI ratios in DO mice have non-genetic contributions outside of XCE-effects. We have included the following text in the Results section as well as an additional Supplemental Figure 10 for this data.

Lines 366 – 389:

“Putative mouse XCE-*Xist*-haplotypes exhibit highly variable XCI ratios

One of the most well-documented instances of a genetic association with XCI ratios is the XCE-haplotypes in laboratory mouse strains, where a preferential ordering of allelic

inactivation exists across haplotypes³³. The DO mice we utilize are expected to be genetically diverse combinations of various lab strains and it is highly likely a mix of XCE-haplotypes are present within this population and may have an impact on XCI ratios. Such a haplotype-specific effect would be missed in our previous AUROC variant-specific analysis of XCI ratios. Since we only sample variants present within RNA molecules and the XCE-interval is a proximal non-transcribed regulatory element of *Xist*³³, we reason variants present within *Xist* are likely linked to XCE-haplotypes and may be informative for identifying putative XCE-*Xist*-haplotypes. We identify 4 putative XCE-*Xist*-haplotypes as determined by groups of samples with shared *Xist* variants (Supp. Fig. 10). As a general observation across haplotypes and the two studies we sample from, XCI ratios of samples with the same haplotype are highly variable (Supp. Fig. 10), suggesting XCI ratios are not definitively determined by *Xist* genotypes within the DO mice population. The haplotype with the seemingly largest effect, evidenced by 2 samples with highly skewed XCI ratios (0.799 and 0.782) in the one study that collected striatal tissue, conversely exhibits XCI ratios ranging from balanced to moderately skewed in the second study, which collected pancreatic tissue (Supp. Fig. 10). While this may be indicative of a tissue-specific effect of a particular XCE-*Xist*-haplotype, far greater sample sizes with higher genetic resolution to confirm haplotypes are needed for validation. In general, the variability in XCI ratios within putative XCE-*Xist*-haplotypes suggests non-genetic contributions to XCI variability of DO mice.”

While the statistical analysis are generally very clear, the notions of “folded” and “unfolded” distribution might be hard to understand for most scientists in the field of XCI who are not highly versed in statistics. The authors could add a few sentences explaining in layman's terms what this represents and use more descriptive terms when describing their figures.

For example, the sentence “we then generate population-level distributions by unfolding the 164 distribution of folded XCI ratio sample estimates per species” line 164-5 might be difficult to understand by some readers.

We thank the reviewer for this suggestion and have included the analogy of opening and closing a book to describe folded and unfolded distributions.

Lines 135–137: “Analogous to folding a book on its side closed, we fold the distribution of reference allelic-expression ratios around 0.50 so that values an equal amount above and below 0.5 are in the same bin.”

Lines 178-181: “Following sample-level XCI ratio modeling, we then generate population-level distributions by unfolding the distribution of folded XCI ratio estimates around 0.50, analogous to opening a closed book (Fig. 1D).”

In figure 1A, the authors fit a distribution to “the tail of the empirical distribution”. While the “tail” are indicated in colors, the author should clearly define the value of the thresholds for the tail (line 208 “XCI ratio estimates between 0.5-0.6, which translates to unfolded estimates between 208 0.4-0.6.

” are 0.4-0.6 the thresholds used? If so this could be mentioned more clearly).

Line 206 The authors also mention “We focus on the tails of the distributions, as our previous validation using phased data indicated” the publication for the “previous validation” should be cited here.

We agree the language can be clearer when describing fitting to the tails of the population distributions and have edited the Results text as follows, now including the proper citation:

Lines 219 – 228: “We fit normal distributions as continuous approximations to the underlying binomial distribution that defines the relationship between cell counts and XCI ratio variability (Fig. 1A, D, see methods). We focus on the tails of the distributions for our model fitting (colored in portions of the distributions, unfolded estimates ≤ 0.40 and ≥ 0.60 , Fig. 2A), for two reasons. Our analysis of autosomal allelic ratios (Supp. Fig. 5) highlights that samples with no expected allelic imbalance produce folded skew estimates that vary between 0.5 and 0.6 and our previous work²⁸ using phased data indicated model misspecification around the point of folding (0.50). Fitting to the tails of the empirical distribution is therefore a more accurate representation of variability specific to XCI.”

In supplementary figure 4, it is unclear what is the value of the x axis “chrX estimated skew”. The authors should clarify what is exactly the value shown here.

Line 220 the author states “The reported X-linked variability in macaca is in excess to the reported autosomal allelic variability ” It is a bit unclear what the authors mean by that and what data support that claim. Are the author referring to the fact that in the 2 plots on top right in Sup Fig 4A, the chrX estimated skew is very slightly higher than the autosomal imbalance in some samples (subsets of dots in the plot)?

We have changed the x-axis label to “XCI ratio” to make it clearer the x-axis is reporting the estimated XCI ratio per sample. For the previous line 220, we are referencing the fact that the Macaca XCI ratios vary slightly higher than the autosomal ratios. We acknowledge this signal is very subtle in the Macaca population. However, we reason that the high consistency of autosomal ratio variation across the species, specifically ratios between 0.50 and 0.60, provides a clear threshold for ratios under the null expectation of balanced biallelic expression. Any ratio greater than 0.60 can be attributed to means other than balanced allelic expression, where in the case of the X-chromosome we attribute it as variation sourced from XCI. We have edited the Results text as follows to make this clearer:

Lines 246-253: “For the least and most variable species (macaca and dog), the estimated autosomal imbalances offer additional context for the reported XCI population variability. The reported X-linked variability in macaca is in excess to the reported autosomal allelic variability, which itself is highly consistent across species (Supp. Fig. 5). This demonstrates the X-linked population variability for macaca, while strikingly small, still varies beyond the extremely consistent autosomal variability present across species and is specific to the X-chromosome, representing informative variability for estimating cell counts.”

In figure 2A , most distributions show a dip around 0.5. Do the authors have an explanation for that?

This is largely due to two reasons: variability in allelic expression across SNPs for a single sample and the model misspecification about the point of folding that we mention for the folded-normal model. Due to a mix of biological and technical reasons, allelic ratios across SNPs for a sample are expected to vary, such as due to sequencing depth or eQTL effects. It is highly unlikely to result in an XCI ratio of exactly 0.50 when aggregating data across numerous noisy SNPs. Such expected variation about 0.50 is heightened when folding a distribution about 0.50; for example, take a sample with SNP allelic expression that varies between 0.4 and 0.6 with a mean of 0.5. When the allelic ratios are folded, the ratios now vary between 0.5 and 0.6 and the mean estimate is likely deviated from 0.50. Our estimates of allelic-imbalances for the autosomes provide a clear example, where the expectation for autosomes is completely balanced allelic expression, yet our estimates of autosomal allelic expression range from 0.5 to 0.6 (Supp. Figure. 5). This is related to why we fit to the tails of the population XCI distributions, to simply exclude the variation immediately about 0.50 that we know is confounded with technical variability and model misspecification.

Fig2 B second panel : to be easier to interpret, the X axis should be labelled in non log scale (absolute number of cells) and the ticks can be log scaled.

We compared the plots using log2 and non-log X-axes, included below, and we still favor using the log2 scale on the X-axis. The point we would like to make is how similar the cell counts are in terms of cell divisions, which we think is clearer when using the log2 scale on the X-axis.